

# Evaluation of Aerosol-Cloud Interactions in E3SM using a Lagrangian Framework

Matthew W. Christensen[1], Po-Lun Ma[1], Peng Wu[1], Adam C. Varble[1], Johannes Mülmenstädt[1], and Jerome D. Fast[1]

[1]Atmospheric Science & Global Change Division, Pacific Northwest National Laboratory, Richland, WA 99354, Washington, USA

**Correspondence:** Matthew Christensen (matt.christensen@pnnl.gov)

**Abstract.** A Lagrangian framework is used to evaluate aerosol-cloud interactions in the U.S. Department of Energy's Energy Exascale Earth System Model (E3SM) version 1 (E3SMv1) for measurements taken at Graciosa Island in the Azores where a U.S. Department of Energy Atmosphere Radiation Measurement (ARM) site is located. This framework uses direct measurements of CCN (instead of relying on satellite retrievals of aerosol optical depth) and incorporates a suite of ground-based

ARM measurements, satellite retrievals, and meteorological reanalysis products that when applied to over a 1,500 trajectories provides key insights into the evolution of low-level clouds and aerosol radiative forcing that is not feasible from a traditional Eulerian analysis framework. Significantly lower concentrations (40%) of surface cloud condensation nuclei (CCN) are measured when precipitation rates in 48-hour back trajectories average above 1.2 mm/d in the Integrated Multi-satellitE Retrievalsfor GPM (IMERG) product. The depletion of CCN when precipitation rates are elevated is nearly twice as large

in the ARM observations compared to E3SMv1 simulations. The model CCN bias remains significant despite modifying the autoconversion and accretion rates in warm clouds.

As the clouds in trajectories associated with larger surface-based CCN advect away from Graciosa Island they maintain higher values of droplet number concentrations ($N_d$) over multiple days in observations and E3SM simulations compared to trajectories that start with lower CCN concentrations. The response remains robust even after controlling for meteorological

factors such as lower troposphere stability, the degree of cloud coupling with the surface, and island wake effects. E3SMv1 simulates a multi-day aerosol effect on clouds and a Twomey radiative effect that is within 30% of the ARM and satellite observations. However, the mean cloud droplet concentration is more than 2-3 times larger than in the observations. While Twomey radiative effects are similar amongst autoconversion and accretion sensitivity experiments the liquid water path and cloud fraction adjustments are positive when using a regression model as opposed to negative when using the present-day

minus pre-industrial day aerosol emissions approach. This result suggests that tuning the autoconversion and accretion alone are unlikely to produce the desired aerosol susceptibilities in E3SMv1.



## 1 Introduction

Microphysical cloud properties, such as the size of cloud droplets and their concentrations, have been shown to change in response to an increase in cloud condensation nuclei (CCN; Twomey, 1974). Increases in CCN lead to predictable increases in droplet number concentration ($N_d$) and smaller cloud droplet effective radii ($R_e$) but only under constant changes in liquid water path (LWP) and cloud fraction (CF). While the Twomey effect is a useful theoretical construct, real-world cloud responses are not instantaneous in time because increased CCN can affect LWP and CF through suppressing or enhancing precipitation and evaporation on timescales from hours to days (Yamaguchi et al., 2017). LWP and CF strongly influence cloud albedo (Stephens et al., 1991) and hence, aerosol indirect radiative forcing, but LWP and CF radiative adjustments (to the Twomey effect) are considered highly uncertain due to the nature in which they co-evolve with precipitation, evaporation, and cloud lifetime (Bellouin et al., 2020). Links amongst cloud macrophysical variables to changes in aerosol processes are poorly constrained in general circulation models (GCMs), typically with unrealistically large LWP responses to increased aerosol concentration (Mülmenstädt et al., 2019; Gryspeerdt et al., 2020). Subtle changes to the tuning parameters of the warm rain process in GCMs to close this gap with observations can result in significant departures in the global mean temperature response due to anthropogenic aerosol (Wang et al., 2012; Golaz et al., 2013, 2019; Mülmenstädt et al., 2020). We thus seek to apply a Lagrangian framework to determine whether macrophysical cloud variables vary over time and as a function of CCN as a means to constrain the aerosol indirect radiative forcing in GCMs and other atmosphere models.

A Lagrangian framework has been demonstrated to be a useful tool to quantify the radiative budgets and impacts of meteorological and aerosol drivers on evolving cloud fields from Large Eddy Simulations (LES; Yamaguchi et al., 2017; Kazil et al., 2021), satellite observations (Pincus et al., 1997; Eastman et al., 2016; Christensen et al., 2020), and aircraft observations (Johnson et al., 2000; Mohrmann et al., 2019). Trajectory analysis is useful for determining aerosol source regions as well as for tracking cloud systems and rates of change in geophysical quantities needed to assess causality. This approach differs from the more common Eulerian perspective which typically lacks key temporal connections between the current cloud state, its prior history, and future changes of precipitation and radiative properties over time. From a Eulerian perspective, aerosol-cloud relationships are typically inferred from a distribution of observations typically taken over many stages of the cloud life-cycle. The extent to whether there is a distinct advantage to using a Lagrangian framework to quantify aerosol-cloud radiation interactions and the impact of cumulative precipitation on aerosol CCN populations forms one of the key motivations for this research.

Routine ground-based observations, such as those taken from the Atmosphere Radiation Measurement (ARM) user facility can complement satellite observations by providing: 1) in situ CCN chamber measurements, 2) vertical profiles of aerosol layers through ground-based lidar observations, 3) vertical estimate of $R_e$, vertical velocity, and turbulent kinetic energy from remote sensing instrumentation, and 4) vertical profiles of atmospheric state properties from radiosondes. These detailed aerosol, cloud, and thermodynamic measurements have been shown to be essential to validate satellite retrieval products (Liu and Li, 2014) and improve process-scale understanding of cloud processes (Wu et al., 2020) at several ARM sites. However, a key limitation is that ground-based measurements can provide only Eulerian perspective. Another goal of this work is to combine





these ground-based data sets within a Lagrangian framework to determine whether warm clouds evolve differently in satellite observations under varying levels of measured CCN concentration. This integrated approach is then used to make the same Lagrangian framework-based comparisons with simulations from the Energy Exascale Earth System Model (E3SM) model.

Recently, Christensen et al. (2020) used geostationary satellite observations in a Lagrangian framework to show that above
median aerosol optical depth (retrieved from confident clear-sky regions) enhances the longevity of marine stratocumulus by about 2 hours along the classic stratus-to-cumulus transition zone. Similar aerosol effects on longevity have been observed and simulated in midlatitude boundary layer clouds (Goren et al., 2019). However, a key limitation of the multi-spectral imagery from polar orbiting and geostationary satellites is that retrievals are typically weighted towards the tops of clouds and thereby unable to inform processes deeper in the cloud (e.g., precipitation). Aerosol optical depth (AOD) or aerosol index ($AI =$Å$\tau_a$;
where Å is the angstrom exponent and $\tau_a$ is the AOD) is commonly used as a proxy for CCN. Aerosol properties cannot be retrieved inside or below clouds using this satellite data and thus, retrievals in clear-sky regions are commonly extrapolated to the observations of nearby clouds. One disadvantage to this lack of collocation (between clouds and clear-sky regions) is that the clear-sky retrievals can be affected by variations in surface albedo, humidification by elevated relative humidity, and 3D radiative scattering off the sides of clouds that artificially illuminate the clear-sky region with the aerosol (Christensen et al.,
2017). Furthermore, AOD and AI are vertically integrated quantities and thus may not represent the CCN concentration at cloud base (Quaas et al., 2020). The lack of vertical co-location between retrieved AOD or AI and cloud-base CCN typically leads to an underestimate in the $N_d$-CCN relationship (Costantino and Breon, 2010). Therefore, it may be prudent to use the CCN (preferably at cloud base) to inform processes related to aerosol-cloud interactions and avoid retrieval issues and assumptions when using AOD or AI from satellite data. To the extent the vertical distribution of aerosol optical depth is correlated to surface-
based CCN and whether it can be used as a suitable proxy for quantifying aerosol indirect radiative effects forms another key question of this research.

The outline of the manuscript is as follows: section 2 describes the data sets used in this study, section 3 provides an example of the Lagrangian framework methodology and its applicability to study aerosol-cloud interactions, section 4 discusses the results and finally a summary of the research is provided in section 5.

## 2   Data

We use a diversity of data sets containing ground-based measurements from ARM as summarized in Table 1, satellite observations from geostationary and polar orbits, and GCM simulations from the E3SMv1 global atmosphere model.

### 2.1   Ground-based observations from the ARM Eastern North Atlantic (ENA) site

The ARM ENA site is located on the northeast side of Graciosa island in the Azores archipelago. Ground-based observations
began starting in 2013, after a successful ARM mobile facility (AMF) deployment during 2009-2010 (Wood et al., 2015). Graciosa resides in the boundary of the subtropics and the midlatitudes and experiences meteorological conditions in both regimes, making it an ideal location to study aerosol and warm cloud properties and processes.



CCN chamber measurements are provided using the Droplet Measurement Technologies (DMT) Model 1 CCN counter (Roberts and Nenes, 2005) at multiple set point supersaturations (0, 0.05, 0.1, 0.2, 0.3, 0.5, and 1.0%). The counter com-
pletes a complete cycle through all supersaturations every 30 minutes. Data is provided from 06/22/2016 – 10/28/2020 in the aosccn1colspectra product (Uin et al., 2016). The data in the value-added product (VAP) is filtered to improve the quality by providing output only when stable measurements occur in which fluctuations in the counter measurements remain below 2 standard deviations within a given saturation step.

We utilize the ARM best-estimate cloud radiation VAP dataset (ARMBECLDRAD; Xie et al., 2010; Tang and Xie, 2020)
which provides the total sky cloud fraction by the total sky imager in hourly intervals from 2014 – 2020 at the ENA site. Cloud top heights and low-level cloud fraction are estimated from the active remote sensing of clouds (ARSCL) product (O'Connor et al., 2004; Kollias et al., 2016) using Ka-Band ARM zenith radar (KAZR) from the ARSCLKAZR1KOLLIAS product (Clothiaux et al., 2001). Ceiliometer retrievals are used to obtain cloud base height using the CEIL VAP. Surface temperature and humidity measurements are used to calculate the lifted condensation level (LCL) are provided from the ARM
best-estimate atmospheric measurements (BEATM) VAP (Xie et al., 2010). These data are used to determine the degree of the coupling within the PBL (see section 4). Surface precipitation rate is obtained from a laser optical OTT Particle Size and Velocity (PARSIVEL) disdrometer which measures the instantaneous rainfall rate of water flux from the number of drops in 32 size (0 to 25 mm) and 32 fall velocity categories (0.2 to 20 m/s) falling to the surface. Precipitation rate from the laser disdrometer has a 6% absolute bias with respect to reference gauges over a 1 min sampling interval (Tokay et al., 2014) as
provided in the LDQUANTS VAP product (Hardin et al., 2020) and averaged into 1-hr intervals to match the temporal sampling of our trajectories.

Layer-mean cloud droplet effective radius is retrieved from cloud optical thickness using the multifilter rotating shadowband radiometer (MFRSR) for overcast single-layer liquid-only cloud layers. The retrieval is based on the algorithm developed by Min and Harrison (1996) of atmosphere radiative transfer at 415 nm. If the liquid water path is available from the microwave
radiometer (MWR) then effective radius is also derived; otherwise, a default value of 8 $\mu$m is assumed in the MFRSRCLDOD VAP dataset (Turner et al., 2021). However, we do not include the default values to ensure independent retrievals are used. This criteria occurs less than 30% of the time and ensures that the results are sensitive to the variations in changes in aerosol concentration. This follows from similar assessments of aerosol-cloud interactions using the MFRSR instrument (e.g. see Kim et al., 2003). Aerosol optical depth is obtained using the MFRSR product (Koontz et al., 2013) from the 550 nm wavelength
by retrieving the total extinction of the direct and diffuse solar radiation.

## 2.2   Satellite observations

Geostationary Operational Environmental Satellite (GOES) Advanced Baseline Imager (ABI) data from the National Oceanic Atmospheric Adminstration (NOAA) GOES-R series satellite is used to track cloud systems across vast regions. The satellite imagery provides several channels spanning the visible, near-infrared and far infrared parts of the electromagnetic spectrum.
At a distance of approximately 36,000 km, the imager views roughly 42% of the Earth's surface (a full disc) in 10 min intervals (although with decreasing spatial resolution at higher latitudes). These observations have the remarkable capability to capture





expansive continental-scale regions at spatial resolutions down to 0.5 km (for the 0.64-$\mu$m visible channel and 2 km for the 3.9-$\mu$m and 11-$\mu$m channels) at nadir making it useful for studying the development and decay of cloud fields.

The CERES SYN 1deg1hr Edition 4.1 product (Doelling et al., 2016) provides $R_e$, visible cloud optical thickness, LWP, and top and bottom of atmosphere longwave and shortwave radiative fluxes. The data is gridded to $1° \times 1°$ spatial resolution at hourly intervals from aggregated retrievals from a network of 16 geostationary satellites (e.g. one of them being GOES-R) which have multi-spectral imagery spanning key channels in the visible, near-infrared, and infrared. The NASA Goddard algorithm retrieves cloud properties following the methodology described in Nakajima and King (1989) and cross-calibrates the data with MODIS collection 5.1 to make a consistent comparison. Finally, an additional calibration step is carried out to ensure the radiative budget of the top of the atmosphere modeled with the Fu Liou algorithm (Fu and Liou, 1992) is consistent with the retrieved CERES top of atmosphere radiative fluxes from the single-scanning radiometer. A recent analysis (Hinkelman and Marchand, 2020) has found that the solar and infrared flux estimates have strong seasonal and diurnal variations that still need to be addressed in the CERES SYN product but despite these biases the relative differences between clean and polluted clouds are informative for studying aerosol-cloud interactions. We compare CERES SYN data with the MODIS collection 6 product at 1-km spatial resolution for the cloud product and aerosol optical thickness retrieved in clear-sky 10 km² regions at 550 and 865 nm wavelengths.

Precipitation rates are provided half-hourly on a 0.1-degree grid using the Integrated Multi-satellitE Retrievals for Global Precipitation Measurement (IMERG) product (Huffman et al., 2015). This product integrates passive microwave precipitation estimates from low earth orbiting satellites (Special Sensor Microwave Imager, Tropical Rainfall Measurement Mission, Advanced Microwave Scanning Radiometer, Global Precipitation Measurement) and infrared imagery at geostationary orbit (ABI and Spinning Enhanced Visible and InfraRed Imager) with the final result tuned to radar and rain gauge retrievals. This dataset has weaker sensitivity to the very lightest precipitation rates (typically less than 0.1 mm/hr) as compared to the active radar on CloudSat (Christensen et al., 2013; Skofronick-Jackson et al., 2018) but has the advantage of continuous spatiotemporal coverage that spaceborne radar does not provide. We recognize that light precipitation can be important for shaping the mesoscale structure of clouds (Savic-Jovcic and Stevens, 2008) and this limitation cannot currently be overcome with existing satellite observations. We thus use accumulated precipitation along trajectories spanning multiple days to examine broad-scale changes in cloud and aerosol properties in a Lagrangian framework.

### 2.3 Meteorological data

Reanalysis data from the Modern-Era Retrospective analysis for Research and Applications, version 2 (MERRA-2) (Gelaro et al., 2017) is used to drive the trajectory model (discussed in subsequent sections). Profiles of temperature, humidity, and wind speed are extracted along trajectories using bilinear interpolation in space and time. MERRA-2 data is spatially gridded at 0.5 degree resolution with 72 vertical levels and provided every 3 hours. Aerosols are included through data assimilation of the bias-corrected aerosol optical depth (AOD) from the Advanced very-high-resolution radiometer and MODIS, Multi-angle imaging spectroradiometer AOD over bright surfaces, and the Aerosol Robotic Network (AERONET) AOD (Randles et al., 2017). Lower-tropospheric stability ($LTS = \theta_{700}$ - $\theta_{sfc}$, where $\theta_{700}$ and $\theta_{sfc}$ are the potential temperatures at 700 hPa and



the surface, respectively) and free-tropospheric humidity (FTH, the relative humidity at 850 hPa; the top of the PBL is found to reside below this level $\sim 90\%$ of the time) are shown to influence the liquid water path adjustment of clouds (Chen et al., 2014). We extract these meteorological quantities from MERRA2 profiles of temperature and specific humidity as well the vertical velocity at 500 hPa.

We investigate whether there is a substantial island wake effect caused by wind blowing over the terrain and whether the downstream cloud properties can be isolated from aerosol impacts on clouds. We compute the Froude number, a dimensionless number defined by the ratio of the flow inertia to the external field which is based on the speed–length ratio and can be written as $F_r = \frac{U/h}{N}$, where $h$ is the characteristic height of the mountains in the Azores (in this case $h = 1000$ m to account for the elevation of surrounding nearby islands where the mountain tops can be significantly higher than Graciosa at 375 m), $U$ is the

perpendicular wind speed from the trajectory at the middle of the PBL impinging on the island, and $N$ is the Brunt–Väisälä frequency which can be written as $N = \sqrt{\frac{g}{\theta}\frac{\partial \theta}{\partial z}}$, where g is the gravitational constant assumed to be 9.81 m/s$^2$, $\theta$ is the potential temperature at the surface, and $\frac{\partial \theta}{\partial z}$ is the potential temperature difference over $h$. If the Froude Number is low ($<1$), it is called subcritical and the flow is blocked by the island and the air will not make it over the top and will instead move around the island. If the Froude Number is high ($>1$) then the flow is called supercritical and will flow freely over the island.

## 2.4    Earth System Modeling

The Energy Exascale Earth System Model (E3SM) version 1 (E3SMv1) (Golaz et al., 2019) is a long-term Earth system modeling effort sponsored by the U.S. Department of Energy. The E3SM Atmosphere Model (EAM) version 1 (EAMv1) (Rasch et al., 2019) uses a spectral-element dynamical core (Dennis et al., 2012; Taylor and Fournier, 2010), an updated two-moment cloud microphysics scheme (Gettelman and Morrison, 2015), the Cloud Layers Unified By Binormals (CLUBB) for

cloud macrophysics, turbulence, and shallow convection scheme (Golaz et al., 2002; Larson et al., 2002; Larson and Golaz, 2005; Bogenschutz et al., 2013), a deep convection parameterization (Zhang and McFarlane, 1995) with convective momentum transport (Richter and Rasch, 2008), a dilute plume treatment scheme (Neale et al., 2008), and a four-mode version of the Modal Aerosol Module (MAM4) (Liu et al., 2012, 2016; Wang et al., 2020). We run the EAMv1 model with a grid spacing of $1° \times 1°$ with 72 vertical levels and nudge the atmospheric horizontal winds toward MERRA-2. Output is saved every hour over a

geographic region spanning $60°E – 20°W$ and $20°N – 75°N$ in the vicinity of Graciosa island. The Cloud Feedback Model Intercomparison Project (CFMIP) Observation Simulator Package (COSP) (Bodas-Salcedo et al., 2011; Swales et al., 2018) and MODIS simulator (Pincus et al., 2012) was used to ensure an apples-to-apples comparison between the EAMv1 model and the MODIS satellite retrievals.

In this study, we test the sensitivity of the autoconversion and accretion parameters to which have been shown to have

a large impact on cloud properties and the radiation budget (Golaz et al., 2013). In the Khairoutdinov and Kogan (2000) (hereafter, KK2000) autoconversion scheme, three parameters (a, b, c) define the autoconversion which can be expressed as $P_{auto} = \frac{\partial q_r}{\partial t}\big|_{auto} = cQ_c^a N_d^b$, where $q_r$ is the rainwater mixing ratio, t is time, and $Q_c$ cloud water content. In the original KK2000 formulation, these parameters were derived by fitting a bin microphysics LES run on one oceanic stratocumulus case. In EAMv1, however, a variant of the KK2000 scheme is used (Rasch et al., 2019) and the parameterization is further adjusted





in E3SMv2 (Ma et al., 2022; Golaz et al., 2022) because these parameters are subject to large uncertainty depending on the cloud regime (Wood, 2005; Kogan, 2013). We reduce the dependency of autoconversion to $N_d$ and increase its dependency on $Q_c$. Four different simulations are performed to examine the parameter space of the effects of autoconversion and accretion on the aerosol indirect radiative effect. Table 2 lists the coefficient and exponents in the autoconversion parameterization for each experiment. In addition, we carry out simulations with pre-industrial emissions of aerosols and their precursors using the same

configuration as our control run (labeled as "A0R0") for present day emissions. The differences between the two simulations reveal the effects of anthropogenic aerosols.

## 3 Methodology

The Lagrangian framework is similar in scope to that described in Christensen et al. (2020) except for two notable exceptions that enhance confidence in our assessment of the indirect radiative effect of aerosols studied here: 1) detailed ground-based

measurements of the aerosol, cloud, and meteorological state from at the start of the trajectory, 2) improved characterization of the polluted vs clean state through the use of actual CCN measurements, and 3) initializing trajectories in more diverse meteorological conditions (i.e. when aerosol and cloud measurements coexist instead of using an AOD retrieval as the proxy for CCN which requires clear-sky conditions to be retrieved from satellite observations). The Hybrid Single-Particle Lagrangian Integrated Trajectory (HYSPLIT) (Stein et al., 2015) version 5 model is used to calculate 48-hour back and forward trajectories

using MERRA2 reanalysis data. Trajectories are calculated from the middle of the planetary boundary layer (as computed using HYSPLIT from the profiles of temperature and humidity). Back trajectories are initialized at the Graciosa Island ARM site each day at 10 am local time to coincide right before the Terra (morning at 10:30 am) and Aqua (afternoon at 1:30 pm) MODIS overpass times. Forward trajectories are initialized 48-hours later to coincide with the arriving airmass following the end of the backtrajectory at Graciosa Island. These trajectories are stitched together to form a 96-hour trajectory starting from the tail

of the back trajectory and ending at the tail of the forward trajectory. This method ensures that the airmass moves through the ARM site and that the meteorological and cloud states would start and end in roughly the same phase of the diurnal cycle for each trajectory. While trajectories could be spawned randomly at different times of the day, starting all of them at the same local time reduces some of the complexity related to temporal changes in radiation and precipitation that are influenced by the diurnal cycle.

### 3.1 Product integration

Figure 1a shows the GOES-R visible image with a trajectory that intersects the ARM site at ENA. Locations along the trajectory are displayed at discrete times ($t_0 = -3.2$, $t_1 = -1.6$, $t_2 = -1$, $t_3 = -0.5$, $t_4 = 0$, $t_5 = 0.5$, $t_6 = 1$, $t_7 = 1.6$, $t_8 = 3.2$ hours) relative to the start of the trajectory (the time at which the trajectory intersects the ARM site). Satellite images of the Lagrangian trajectory at these discrete times are displayed in Figure S1. The trajectory tracks well with the boundary layer

clouds as depicted in Movie S1. This particular case was selected to highlight the stratus-to-cumulus transition that sometimes occur in this region. The large bank of closed cell type stratocumulus clouds to the northeast of Graciosa Island and lack of





overlying high-level clouds as indicated by the relatively high values (greater than 273 K) of the 11-$\mu$m brightness temperature (an indicator for the presence of low-level clouds) make this a notable example (Fig. 1b). During the day the cloud bank continuously impinges on the island producing precipitation with Ka-band radar reflectivities as large as 15 dBZ (Movie S2 as observed from a Eulerian perspective).

The $R_e$ retrieved using the 3.7-$\mu$m channel from MODIS is in good agreement (within the noise of the instruments and differences between a layer-mean average and a retrieval near cloud top; in this case about 2 $\mu$m) with that retrieved from the ARM MFRSR instrument (Figure 1c). This is further confirmed through examination of coincident ARM retrievals of $R_e$ from an Eulerian perspective (centered over the ARM site; Figure S2). Figure S2 reveals a bias in the simulated $R_e$ from E3SMv1 using the COSP satellite (observation) simulator. The bias remains despite changing the autoconversion parameter values or from even turning off anthropogenic aerosol emissions (i.e. similar to a pre-industrial based run) altogether. For the first half of the trajectory the cloud fraction is 1.0 (Figure 1c) but shortly after passing the ARM site the cloud fraction decreases in the wake of the island and then recovers to ∼0.6 about 6 hours later. The Froude number was estimated to be 2.5 for this case thus indicating that the flow was not blocked by the islands within the Azores but a wake in the lee of the island is observable in Movie S1. The gradual increase in sea surface temperature over the trajectory (Figure 1d) and cloud clearing possibly from the island wake result in larger observed brightness temperatures downstream from the island where warmer sea surface temperature (SST) occurs. Due to the potential impact of the island topography on the cloud properties we stratify trajectories by a wide range of meteorological indices, including Froude number, in this study.

## 3.2 Screening procedure

A total of 1589 (96-hour) trajectories are calculated for the time period between 6/22/16 – 10/28/20 based on a once-per-day strategy. We use two separate screening approaches to 1) examine the effect of precipitation along **back trajectories** (section 4.1) on measured and simulated CCN and 2) quantify the aerosol indirect radiative effect and meteorological drivers on warm cloud at Graciosa Island in **forward trajectories** (section 4.2 - 4.5). For the first analysis, we consider all precipitation types (shallow and deep convection) occurring in the airmass of all back trajectories and split trajectories into those with relatively low and high median rates of precipitation.

For the second analysis, we restrict the selection criteria to only single-layer warm (cloud top temperature greater than 273 K), low-level (cloud top pressure greater than 500 hPa), liquid phase clouds along both the stitched-together backward and forward trajectories. This screening criteria is essential to avoid uncertainties related to glaciation indirect effects involving aerosol-ice-cloud interactions (Lohmann, 2002). A time-step in the trajectory is ignored if any mixed-phase or ice clouds are detected. If ice cloud is found to occur in more than 25% of the time-steps the whole trajectory is removed from the analysis. Trajectories are split into two categories based on their initial value of CCN. The peak of the CCN distribution at the ARM Graciosa Island site is approximately 110 cm$^{-3}$. Therefore, to ensure roughly a similar number of samples between the distributions, a trajectory is considered clean if CCN < 110 cm$^{-3}$ and polluted if CCN > 110 cm$^{-3}$.





## 4 Results

The spatial distribution of the trajectories of this study are shown in Figure S3. Trajectories mostly originate from the Southwest and translate to the NE over time which is consistent with the dominant wind-flow pattern in this region (Wood et al., 2017).

### 4.1 Back trajectories: Precipitation and CCN

Is the ARM measured CCN concentration affected by precipitation along back trajectories? The CCN budget is largely dictated by aerosol sinks (via dry deposition, wet deposition, and advection), surface sources (e.g., sea-spray production, horizontal and vertical advection, or by new particle formation and growth), and aerosol chemistry in this region (Wood et al., 2017). Figure 2 shows two time-series of CCN from both ARM measurements and E3SM simulations. A seasonal cycle is found in both datasets with larger values of CCN occurring in the summer and lower values in the winter. The minima in winter months are likely associated with more frequent passages of frontal systems and cold air outbreaks producing more precipitation and cleaner atmospheric conditions (Wood et al., 2017).

Figure 3a shows a histogram of the CCN concentration for trajectories sorted by precipitation state. The average accumulated precipitation amount from IMERG and E3SM datasets along 48-hour back trajectories is $2.50 \pm 7.5$ mm and $2.48 \pm 4.68$ mm, respectively. The accumulation totals are about the same between observations and the model but the standard deviation is larger in the IMERG data which denotes a wider range of estimates than E3SM. Heavy-rain and light-rain trajectories are sorted based on this accumulated average threshold of 0.05 mm/hr (depicted in Figure S4) . Figure 3a shows that there is a 40% decrease in the mean CCN concentration at the ARM ENA site when above median precipitation occurs 48 hours prior to the measurement time. In addition, we find that CCN is decreased when above-median precipitation occurs even in an Eulerian framework (Figure S5) but with only a slightly less pronounced change than when using a Lagrangian perspective as compared to the results displayed in Figure 3.

The extent to whether wet deposition is responsible for the reduction in measured CCN or if the difference in CCN can be explained by the advection of different airmasses with different types of aerosols in the back trajectories is examined using the aerosol budgets from the MERRA2 reanalysis data. Evidently, sea salt and sulfate make up the bulk of the AOD in this region (Figure 4a). The change in total AOD over time along trajectories is slightly positive but the changes are broadly similar for each aerosol species. In addition, Figure S6 shows that the difference in AOD between clean and polluted composites is roughly in balance for each aerosol species over the course of the trajectory. This result may imply that the aerosol sinks and sources are also roughly in balance along the trajectories on average and that the changes associated with CCN concentration are may be tied to local-scale disturbances caused by precipitation and not by the advection of changing aerosol concentrations along the trajectories. This is in general agreement with the CCN closure model used by Wood et al. (2017) which found that the CCN is strongly tied to wet deposition of aerosol by the precipitation particularly when clouds have high LWPs.

The CCN concentration typically decreases following the passage of precipitation over the ARM site. This effect is shown in an Eulerian framework using disdrometer precipitation measurements (Figure 5a), and E3SM simulations (Figure 5c). The CCN concentration is linked in time to the presence of precipitation. Although this example shows that CCN gradually de-





creased in the E3SM simulations a couple days before heavy precipitation occurred but then rapidly decreased after rainfall. Figure 5b shows that on average there is a precipitous decline in CCN concentration in the hours prior to the occurrence of precipitation with the largest declines occurring at the time of the rainfall (all prior times have to have either no rainfall or rain-

fall less than 0.05 mm/hr). The observations show a much sharper decline in CCN concentrations as a function of time until rainfall ($-63\%$ using the ARM disdrometer and $-28\%$ using IMERG over 20-h) compared to the E3SM simulations ($-9\%$). At the time of rainfall, CCN concentration decreases as hourly-rain rates increase ($R$) in the ARM observations ($\frac{\Delta CCN}{\Delta R} =$ -24.98 cm$^{-3}$/(mm/hr)) and this rate of change is more negative compared to the E3SM (A0R0) simulations ($\frac{\Delta CCN}{\Delta R} =$ -17.63 cm$^{-3}$/(mm/hr)) (Figure S7).

The lack of a substantial difference in CCN caused by precipitation between Eulerian and Lagrangian perspectives (i.e. differences between Figure 3 and Figure S5) in the observations may be due to insufficient sensitivity in the precipitation detection from the IMERG product as may be indicative of the weaker relationship found when examining this product in an Eulerian framework. Small differences between frameworks could also manifest if the CCN is depleted rapidly in time with respect to the temporal time-step of 1 hour by precipitation. The precipitous drop in CCN in the few hours within the time until

a rain event occurs may indicate that precipitation is very effective at removing CCN from the atmosphere and could explain why Lagrangian and Eulerian frameworks have similar results for this set of particular variables in this location. However, we cannot entirely rule out the effects of shifting air masses, for example the passage of cold fronts that cleanse the airmass before arrival at the ARM ENA site, that may also cause precipitous drops of CCN in an Eulerian framework without detailed CCN measurements along trajectories.

A decrease in CCN is simulated when the trajectories have precipitation rates greater than a threshold of 0.05 mm/hr in the E3SM model (Figure 3b) though the response in E3SM is only about half as large as in the observations. The lack of CCN removal in the model by precipitation may indicate that E3SM does not have strong enough wet deposition at this location for processing/removal of aerosol from the atmosphere. Increasing the accretion efficiency decreases the base-state CCN concentration but not the rate of decline with respect to this timescale to precipitation occurrence (Figure 5d). This

inference is supported by the similar responses in all of the autoconversion/accretion experiments listed in Table 3 and from the relationship between CCN and rain rate (Figure S7). The fractional decrease in CCN by precipitation is two times smaller in the pre-industrial based atmosphere compared to the present day. This result implies that precipitation is more efficient at removing aerosol from the atmosphere in present-day conditions, probably because there is simply more aerosol to remove. Unlike the observations that use IMERG, a Lagrangian perspective enhances the change in CCN between precipitating composites

compared to the Eulerian perspective in the E3SM model. Despite the enhancement in CCN removal by precipitation in the Lagrangian framework, a significant bias remains and deeper investigation into the sources and sinks of the CCN in the E3SM model is outside of the scope of this study but warranted for future releases of this version of the model.

## 4.2 Forward trajectories: Cloud Radiative Effects

Forward trajectories are used to quantify the cloud property perturbations to changes in aerosol concentration downstream
from the ENA site. Trajectories are sorted by above and below median CCN concentrations (defined here as polluted and clean





with a threshold of $110 \, \text{cm}^{-3}$) giving approximately 650 clean and polluted warm-cloud cases to quantify the aerosol response to warm clouds. Coincidentally, the two classes of trajectories flow in similar directions (Fig. S8) and as a consequence, the meteorology is broadly consistent between them both at the ENA site (Fig. S9) and in-time along the trajectories (Fig. S10). While the meteorological means are within their respective ranges of the uncertainties between clean and polluted composites

even small changes, in for example LTS (in this case being lower on average in the clean trajectories), could be relevant for the cloud properties (Klein and Hartmann, 1993) and hence why we bin by meteorological variables to examine the impact of aerosols in section 4.4. Figure 6 generally shows that higher AOD is associated with the more polluted trajectories with larger concentrations of CCN. Despite the relatively higher noise in MODIS, AOD remains elevated in the available hourly intervals along trajectories that are polluted (star) compared to clean (circle) samples. While this difference is substantial (albeit less

so in MERRA2 which may suggest deeper investigation into MERRA2's ability represent the evolution of aerosols) the AOD differences tend to be largest at the start of the trajectory (i.e. at t=0) and gradually become smaller over the next 48 hours. As a result, when higher concentrations of CCN are measured at the ARM site, CCN likely remains elevated in the atmosphere for multiple days as inferred by the change in downstream AOD. The extent to whether the elevated aerosol concentrations influence cloud properties and radiative budgets along trajectories are examined next.

### 335 4.3 Aerosol influence on cloud properties

Figure 7a shows that smaller $R_e$ manifests in more polluted conditions (trajectories with higher CCN). This result is consistent with other studies using ARM (Dong et al., 2015), satellite (Christensen et al., 2020), and modeling (Yamaguchi et al., 2017) data. Furthermore, the Lagrangian framework reveals that $R_e$ remains smaller in polluted clouds even 20-30 hours after being sampled at the ARM site. This result is consistent with the elevated AOD over the trajectory and generally agrees with the

central premise of the Twomey effect in which a larger concentration of CCN produces a larger concentration of smaller cloud droplets (the computation of $N_d$ is described in subsection 4.5). Interestingly, the CERES SYN retrievals of $R_e$ remain smaller in polluted clouds even at night (hours 10 - 22 and 32 - 44) when the retrievals rely exclusively on near infrared brightness temperature and are, thus, much less trustworthy. On average, cloud optical thickness and $N_d$ (Figure 7b,c) is larger in clouds (although $N_d$ has a much larger difference than optical depth due to the LWP differences that are opposite) that are polluted compared to those that are unpolluted. LWP and CF (Figure 7d,e) exhibit average decreases at elevated CCN concentration in

most data sets (except at nighttime in CERES data where the retrievals are less trustworthy and in MODIS on the second day along the trajectory). These estimates are also provided in Tables 5 and 7. The decrease in liquid water path and cloud fraction occur despite the significant suppression of precipitation in polluted conditions on average (Figure 7f); a response which would increase LWP rather than decrease it. A possible explanation for the polluted clouds that lose water despite moistening by

drizzle suppression is that the evaporation may be more vigorous due to differing meteorological factors compared to the clean clouds with more liquid water (Chen et al., 2014).



## 4.4 Meteorological factors

We seek to understand why the LWP and CF decrease at elevated CCN concentration by examining the aerosol response to several key geophysical variables: LTS, FTH, 500-hPa vertical velocity, free tropospheric vertical velocity $\omega_{500}$, degree of

cloud coupling to surface moisture, fraction of the aerosol residing in the PBL (compared to the free troposphere), precipitation state, and the Froude number. We compute the degree of decoupling based on the difference in ceilometer cloud base height and lifted condensation level (computed using radiosonde observations with a parcel height level of approximately 100 m above the surface at 10 am local time). If the distance is less than 300 m we consider the PBL to be well-mixed with higher likelihood of surface aerosol mixing with overlying clouds according to (Comstock et al., 2005). An example of the method is displayed in

Figure S11. We find that the selection of warm clouds at ENA are more often decoupled to the surface moisture supply (about 54%); Rémillard et al. (2012) found that despite a high frequency of stratocumulus clouds in the Azores, the PBL is almost never well mixed but is often in a meteorological state where the cumulus clouds are coupled to the surface moisture supply.

Does the vertical distribution of AOD and its location in the vertical influence cloud properties? Since aerosol can sometimes reside above the cloud tops and be poorly collocated with the clouds, we extracted the vertical profile of aerosol optical depth

along trajectories using the MERRA2 data. Aerosol optical depth was computed using look up tables for the dry mass extinction coefficients for each aerosol species (see supplementary materials Table 1 in Randles et al., 2017). The mean PBL height as determined from MERRA2 is 912 hPa at the ENA ARM site. We estimate that about 55% of the daily 12 UTC time column integrated AOD is located within the PBL (Figure 4b). Because an appreciable amount of the aerosol loading can reside above the PBL we quantify cloud property changes in response to changes in CCN separately for cases in which a majority of the

aerosol resides in the PBL versus times when it does not as determined by MERRA2.

Does the island create a geophysical wake and influence the cloud properties in Lagrangian trajectories thereby obscuring the aerosol-cloud relationship? The interactions of the incoming winds and high mountainous areas induces a wind-sheltered area and wake effect in the lee of the island. Figure S12 clearly shows a strong decrease in cloud fraction on average in the wake of the Madeira islands - this is consistent with observations from (Azevedo et al., 2021). It is also evident that LWP

decreases in the wake of Madeira and cloud droplet concentration remains constant. On the other hand, we do not see strong signatures of an island wake effect on the clouds downstream from ENA. Instead we do observe local anomalies in the cloud properties at ENA but these are on relatively short timescales (less than 30 minutes) and likely a retrieval artefact caused by surface inhomegenieties resulting from the enhanced reflectance over land compared to the ocean surface (Fig. S12b). Cloud properties are also examined as a function of Froude number (Fig. S13); however, changing the Froude number does not

strongly influence the cloud properties above the statistical noise in the region. Therefore, we conclude that while the wake generated by Graciosa and nearby islands in the Azores may influence clouds on specific days (e.g., Figure 1a) it does not significantly influence the cloud properties in a multi-year average.

Taken together, Figure 8 shows the average difference in $N_d$ between clean and polluted clouds that have been stratified by low and high value thresholds of a variety of different meteorological composites. In general, $N_d$ is larger in trajectories with

higher CCN (i.e. a positive $\Delta N_d$ value) and the differences amongst meteorological composites are not statistically different



except for some notable exceptions. Higher $N_d$ tends to be measured in polluted clouds when the atmospheric conditions tend to be more stable (larger $\Delta N_d$ when the LTS is larger than the median conditions) and moist (using FTH). The variations in $\Delta N_d$ across meteorological composites are slightly larger for ARM than for MODIS. ARM measurements indicate that significantly less $\Delta N_d$ is measured in polluted clouds that are precipitating but these retrievals may be affected by sub-cloud rain influencing

the cloud retrieval (Wu et al., 2020) whereas rain contamination on the passive near-infrared cloud top retrievals from MODIS is smaller (Christensen et al., 2013). Overall, we observe broad agreement on the sign of $\Delta N_d$ amongst MODIS, ARM, and E3SM but the strength and sensitivity to environmental factors can vary significantly between them.

The change in LWP (Fig. S14) and CF (Fig. S15) is more likely to become positive if the free troposphere is moist, the clouds are precipitating, or the Froude number is large, as indicated by the MODIS results. These result generally agree with

the hypothesis that enhanced entrainment of dry air into polluted clouds causes them to lose more LWP (Chen et al., 2014) and that these responses are more likely to occur during summer months (not shown) when the atmosphere is drier compare to winter months (Dong et al., 2015; Wood et al., 2015). Strong positive $\Delta LWP$ in raining clouds is also in agreement with the CloudSat observations from (Chen et al., 2014) and is hypothesized to result from the suppression of precipitation, which acts as a cloud moisture sink. There is also some evidence that for clouds that are less coupled to surface moisture, or if more of

the aerosol is in the free troposphere, greater losses in LWP and CF may occur when the clouds become polluted. Decoupled clouds have been shown to be more susceptible to greater losses in LWP when they become polluted because there is less ability to transport moisture from near the surface and replenish the cloud (Zheng et al., 2022). While the sign of the responses from ARM are sometimes different from MODIS, the relative change between high and low for each of the meteorological composites is in broad agreement with MODIS and less so with E3SM simulations (e.g. $\Delta LWP$ and $\Delta CF_{liq}$ are mostly

negative across meteorological composites in E3SM but have much bigger variations in ARM and MODIS by comparison).

### 4.5   Effective radiative forcing by aerosol-cloud interactions ($ERF_{aci}$)

The radiative effect due to changes in aerosol concentration is decomposed into contributions from the Twomey effect and by adjustments caused by changes in LWP and CF. We use a bivariate statistics approach and split the data into "clean" and "polluted" states based on two methods: 1) from present-day composites of the data separated by CCN concentration and

2) from present-day ("polluted") and pre-industrial based ("clean") conditions. Obviously, we can only assess $ERF_{aci}$ from satellite observations from the first approach, whereas the GCM is able to compute $ERF_{aci}$ using both approaches. Running a pre-industrial based simulation also provides the needed constraint for determining annual mean incoming TOA shortwave radiation and anthropogenic aerosol fraction (to be used in approach 1).

### 4.5.1   $ERF_{aci}$ Derivation

The change in reflected solar radiation caused by a change in the planetary albedo and $N_d$ can be written as

$$ERFaci = \overline{F^\downarrow}\Delta\alpha = \overline{F^\downarrow}\frac{d\alpha}{dN_d}\overline{\Delta N_d} \qquad (1)$$





where, $\overline{F^\downarrow}$ is the annual mean top of atmosphere (TOA) incoming solar radiation taking a value of 329 W m$^{-2}$ (is the daily mean annually averaged value at Graciosa Island), $\alpha$ is the planetary albedo, $\overline{\Delta N_d}$ is the change in droplet concentration due to anthropogenic activities which can be approximated from the change in aerosol optical thickness (Quaas et al., 2008) from

pre-industrial to present day levels via $\overline{\Delta N_d} = \Delta N_d \frac{\tau_a^{PD} - \tau_a^{PI}}{\Delta \tau_a}$, where $\Delta N_d$ and $\Delta \tau_a$ are the changes in droplet concentration and aerosol optical thickness between clean and polluted conditions, respectively. $\alpha$ can be expanded into contributions from the surface and clouds following

$$\alpha = (1 - f_c)\alpha_{clr}\phi_{atm} + \alpha_c\phi_{atm}f_c \tag{2}$$

where $\phi_{atm}$ is the transfer function that accounts for the average albedo of the air above the surface and clouds and takes

an average value of 0.7 (Diamond et al., 2020), $\alpha_c$ can be estimated using the two-stream delta Eddington approximation assuming the surface albedo beneath the cloud is zero as

$$\alpha_c = \frac{(1 - g)\tau_c}{2 + (1 - g)\tau_c} \tag{3}$$

where, $g$ is the asymmetry parameter and takes a value of 0.85 for liquid clouds and $\tau_c$ is the cloud optical thickness which is approximated using an adiabatic assumption as $\tau_c = \gamma^p L^{\frac{5}{6}} N_d^{\frac{1}{3}}$ where $\gamma^p$ is a constant value of 1.37e-5 m$^{-0.5}$, $L$ is the LWP

and $N_d$ is the cloud droplet concentration. Taking the derivative of $\alpha$ with respect to $N_d$ gives

$$\frac{d\alpha}{dN_d} = \phi_{atm}\left(-\alpha_{clr}\frac{\partial f_c}{\partial N_d} + \alpha_c\frac{\partial f_c}{\partial N_d} + f_c\frac{\partial \alpha_c}{\partial N_d}\right) \tag{4}$$

where cloud-free conditions give $\frac{\partial \alpha_{clr}}{\partial N_d} = 0$. The chain rule expansion of $\frac{d\alpha_c}{dN_d} = \frac{\partial \tau_c}{\partial N_d}\frac{\partial \alpha_c}{\partial \tau_c}$ can be solved by the following two derivatives: 1) $\frac{\partial \tau_c}{\partial N_d} = \frac{\tau_c}{3N_d}\left(1 + \frac{5}{2}\frac{\partial \ln N_d}{\partial \ln L}\right)$ and 2) $\frac{\partial \alpha_c}{\partial \tau_c} = \frac{\alpha_c(1-\alpha_c)}{\tau_c}$. Combining with equation (3) gives the resulting equation

$$ERF_{aci} = -F^\downarrow \phi_{atm}\frac{f_c\alpha_c(1-\alpha_c)}{3N_d}\left(1 + \frac{5}{2}\frac{\Delta \ln L}{\Delta \ln N_d} + \frac{3(\alpha_c - \alpha_{clr})}{\alpha_c(1-\alpha_c)}\frac{\Delta \ln f_c}{\Delta \ln N_d}\right)\Delta N_d\frac{\tau_a^{PD} - \tau_a^{PI}}{\Delta \tau_a} \tag{5}$$

which is used to compute the aerosol indirect shortwave radiative effect. Here, single-directional difference quotients $((\Delta Y/\Delta X)|_Z \approx \partial Y/\partial X)$ are represented as a linear relationship, however, they depend upon meteorological conditions and the background aerosol conditions (Glassmeier et al., 2019).

### 4.5.2 Present-day $ERF_{aci}$ based on CCN subsets ($1^{st}$ Approach)

The $\Delta$ terms represent differences in cloud properties between the clean and polluted state based on measured values of the

CCN at the ARM site. The differences are computed between the clean and polluted forward trajectories at each timestep. The first term in brackets is commonly referred to as the Twomey effect denoting the change in shortwave reflection assuming





liquid water path and cloud fraction terms (the following two terms) are zero. Liquid water path and cloud fraction changes are commonly referred to as radiative adjustments of the Twomey effect (IPCC, 2013).

Radiative effect estimates from trajectories intersecting the ENA site are displayed in Table 4 and associated cloud quantities

are listed in Table 5. The Twomey radiative effect is negative in all data sets and spans the range from $-0.72$ to $-1.82$ W m$^{-2}$. This is a wider range of estimates compared with global-based observational and model-based estimates (e.g. see, Quaas et al., 2008; Lebsock et al., 2008; Christensen et al., 2017), and may be due to sampling a particular cloud environmental regime that is more susceptible to aerosol perturbations than the aggregated global mean value; further analysis contrasting other sites and regions will be examined in follow up work.

The Twomey effect is more negative in the E3SM model and ARM observations (averaged over the first day of the lagrangian trajectory), in part, because these data sets exhibit larger increases in $\Delta \ln N_d$ under polluted conditions (i.e. larger values of $\Delta \ln N_d$) compared to the retrievals from the satellite observations. The Twomey effect is also inversely proportional to $N_d$, thus, larger values of $N_d$ result in smaller radiative effects. Part of the reason why the Twomey effect in ARM data is much more negative than the satellite estimated values is because the cloud albedo and $N_d$ is nearly two times larger. Similarly, the

mean $N_d$ in the E3SM model is significantly larger than in the observations. Despite the relatively large $N_d$ in ARM, E3SM remains an outlier with respect to the observations of $N_d$ and provides estimates of the Twomey radiative effect to within 30% of the observations.

The modification of the autoconversion and accretion parameterizations do not significantly influence the Twomey radiative effect (Table 4 and Table 6) or base-state cloud variables (i.e. cloud fraction, cloud albedo, and droplet number concentration;

Table 5 and Table 7). However, the LWP radiative adjustment becomes smaller between autoconversion experiments (experiment A1 compared to A0) and this is consistent with stronger precipitation suppression when the dependence of precipitation is greater with larger $N_d$ (also shown in prior studies; Gettelman et al., 2021). Overall, the modification of these autoconversion and accretion factors have a small influence on the time-series of the CCN (not shown) and $N_d$ (Figure 9a) as well as the aerosol indirect radiative effect (Table 4) over this parameter space. A much larger and more pronounced effect on these vari-

ables occurs when the anthropogenic aerosol emissions are turned off in the model (A0R0PI simulation). This result implies that despite recent upgrades to the E3SM code base (Wang et al., 2020) there remains a relatively weak connection between the warm rain process, specifically autoconversion and accretion, and cloud radiative effects in the model.

The ENA site shows a negative LWP response to increasing aerosol concentrations in E3SM and observational datasets. This leads to a positive radiative effect despite the large increase in $\Delta N_d$ (Figure 9a) and outweighs the cooling caused by

the Twomey radiative effect. The negative adjustments are likely driven by the strong descending branch in the $LWP - N_d$ relationship (Figure 9b). This effect is robust in the pre-industrial and all present-day autoconversion and accretion experiments. The descending branch may be attributed to enhanced entrainment drying on clouds as they become increasingly polluted (Grysspeerdt et al., 2019). While the mechanism for entrainment drying is included in E3SMv1 via the connection between aerosols and droplet sedimentation (Bretherton et al., 2007) we speculate that the droplet population may not have enough time

to separate vertically over the integration timescale (Karset et al., 2020) and therefore we do not expect the model to simulate negative slopes for LWP as a function of $N_d$ (and yet it does). Thus, further developments on improving the biases with respect





to CCN concentration, precipitation rates, and $N_d$ may be needed to bring radiative forcing estimates in better agreement with the observations.

The net indirect radiative effect on the first day along the trajectories takes on a positive value in E3SM simulations despite the negative contributions from the Twomey effect. The positive warming effect manifests from the decrease in both LWP and CF as CCN levels increase. These adjustments are nearly as large in magnitude as the Twomey effect in the E3SM model. As a consequence, the net indirect radiative effect takes on a positive value after these contributions are added to the Twomey effect. The satellite and ground-based measurements also show positive LWP and CF adjustments; however, their contributions are much smaller by comparison to the Twomey effect. As a result, the net indirect radiative effect remains negative.

Cloud responses to changes in CCN concentration also change between days 1 and 2 along the trajectories (Tables 5 & 7). While the $\Delta \ln N_d$ remains positive on day 2, it decreased by 32% in MODIS observations and 15% in E3SM simulations. The LWP and CF adjustments are less consistent in their change between days. MODIS observations show a reversal of the sign on the second day such that polluted clouds become associated with larger cloud fraction and liquid water paths. This result could imply that microphysical changes associated with increased aerosol loading are less affected over time compared 490 to macrophysical cloud properties such as LWP and CF which may be more affected by meteorology, precipitation, and evaporation processes where $N_d$ may be less impacted because it is considered a mediating variable that is less directly influenced by relative humidity (Gryspeerdt et al., 2016). Another potential caveat is the influence of the diurnal cycle on the cloud properties. LWPs are more decreased in polluted clouds during the afternoon and evening hours, a result that broadly agrees with observations of polluted cloud tracks (Rahu et al., 2022). The cloud fraction is decreased in polluted clouds the most during 495 the night into early morning hours in the E3SM model.

### 4.5.3    Present-day minus Pre-industrial $ERF_{aci}$ ($2^{nd}$ Approach)

Here, the $\Delta$ terms in equation 5 represent differences in cloud properties between the present-day and pre-industrial based atmosphere from E3SM. Table 8 shows some notable differences between using the bivariate statistics regression approach using $\Delta$CCN ($1^{st}$ approach) and $\Delta$(PD-PI) in computing $ERF_{aci}$. Firstly, $N_d$ is significantly larger, $R_e$ is significantly smaller, 500 and cloud optical thickness is larger with present-day emissions. This agrees with our first approach and the general Twomey hypothesis. Interestingly, LWP increases from pre-industrial to present-day, not decreases as shown using the bivariate statistics of $\Delta$CCN. As a consequence, this causes a stronger negative net radiative cooling effect. Similarly, CF increases, thereby leading to additional cooling, not warming as before.

     The discrepancy between the bivariate statistical approaches may result from a variety of factors (e.g. see Ghan et al., 2016; 505 Bellouin et al., 2020) broadly summarized here as: 1) co-variability between meteorology and the cloud variables may affect the sensitivities in $\frac{\Delta \ln LWP}{\Delta \ln N_d}$ and $\frac{\Delta \ln CF}{\Delta \ln N_d}$ using the $\Delta$CCN approach, 2) assumptions in equation 5 ($\phi_{atm} = 0.7$, $g = 0.85$, and the adiabatic approximation for estimating LWP and $N_d$, to name a few) may not be reasonable for certain types of clouds, such as those where the cloud fraction is low (Coakley et al., 2005), decoupled from the surface, or precipitating, 3) the estimate of the pre-industrial aerosol state is poorly constrained and uncertain (Carslaw et al., 2013), and 4) shifts in cloud properties in 510 the pre-industrial based simulation has smaller $N_d$ but with decreased LWP and CF effects relative to Twomey (see Table 4).





We compute a residual term in the calculation of $ERF_{aci}$ of $0.37 \pm 0.18$ W/m$^{-2}$ as determined from the difference between $ERF_{aci}$ and $\Delta \mathrm{F}_{SW}^{\uparrow}$ in Table 8. The size of the residual suggests that this approach captures nearly most of the variability and that the decomposition of $ERF_{aci}$ is accurate to within 85%. Based on this level of accuracy we are confident in the estimates of the Twomey effect but less confident that the changes in LWP and CF are statistically different from zero. We

aim to determine which of these factors is most responsible for the computed residual and differences between the bivariate statistics for $\Delta$CCN and $\Delta$(PD-PI) in computing $ERF_{aci}$ in a follow up study.

## 5    Conclusions

This study utilizes a Lagrangian framework to characterize the radiative effect of aerosols on clouds passing by the ARM site at Graciosa Island. This framework is applied to over 1,500 trajectories which track warm boundary layer clouds at dis-

tances of over several hundred kilometers in both satellite observations as well as in GCM simulations. This approach utilizes ground-based measurements of CCN instead of satellite retrieved AOD thus bypassing the need to rely exclusively on clear-sky conditions to initialize trajectories. Below we summarize answers to the key scientific questions of this study:

1. **Is the Lagrangian framework advantageous for quantifying the relationship between CCN concentration and precipitation occurrence?** In agreement with Wood et al. (2017) we show that CCN concentrations precipitously declines

in association with precipitation occurring along back trajectories. We find increased sensitivity in the E3SM model when using a Lagrangian framework but this relationship was relatively unchanged by the autoconversion or accretion experiments conducted in the E3SM model. An analysis using an Eulerian framework found similar results but with only slightly lower sensitivity than in the satellite observations, possibly due to sensors that are not able to capture light rain.

2. **Do cloud properties evolve differently under varying levels of measured CCN?** Yes, our analysis shows that clouds

tend to have higher $N_d$ at higher starting concentrations of CCN and that $N_d$ and other cloud properties remain perturbed downstream from ENA for several days, albeit with decreasing strength over time. E3SMv1 is able to capture the temporal response in $\Delta N_d$ but the decrease over time is weaker than the observations would otherwise suggest. These perturbations result in aerosol indirect radiative effects. While the microphysical changes in the cloud lead to a cooling Twomey radiative effect, we also find substantial warming radiative effects by decreases in LWP and CF that are ob-

served in satellite observations as well as simulated with larger estimates in the E3SM model. The positive LWP and CF adjustments may be the result of the bivariate statistical sampling of clean and polluted samples occuring on the decending branch of the $LWP - N_d$ relationship (Figure 9b). E3SM simulations show that LWP and CF actually increase from the pre-industrial to present-day aerosol emissions which causes even more radiative cooling using the PD-PI approach. The lack of agreement between the bivariate regression and PD-PI sampling approaches is large with a complete reversal

of $ERF_{aci}$ suggesting that further assessments of the Twomey, LWP and CF decomposition calculation which many other studies utilize (Quaas et al., 2008; Goren and Rosenfeld, 2015; Bellouin et al., 2013; Christensen et al., 2017; Toll et al., 2019; Christensen et al., 2020) are needed to assess factors that may violate assumptions using these approaches.



3. **Is the Lagrangian flow affected by Graciosa Island?** No, we do not find strong evidence of the flow affecting the statistical relationships between CCN concentration and cloud properties when averaged across multiple years. Although, an island wake effect can be observed in individual case studies (e.g. Figure 1), analysis of the Froude number suggests blocking is relatively rare and the influence on cloud properties is not statistically different from zero.

4. **Is the vertical distribution of AOD correlated with surface-based CCN?** We observe increased $N_d$ in overlying clouds for larger values of surface-measured CCN regardless of the degree of surface-to-cloud coupling. $N_d$ remained elevated despite conditions when the bulk of the aerosol resided above the planetary boundary layer (Figure 4b) as determined from the MERRA2 reanalysis product. These responses are robust across a multitude of meteorological parameters and factors that are influenced by the island wake effect.

In general, E3SM simulates positive LWP and CF radiative adjustments (using $\Delta CCN$ as a proxy for bivariate regression method 1 statistics) regardless of the meteorological state. While the differences between E3SM and the observations are notable, the lack of consistency between ARM, MODIS, and CERES makes it difficult to constrain cloud system behavior based on different meteorological states in the simulations. This challenge was also identified in Neubauer et al. (2017) in which the sign of the LWP response from MODIS and Advanced Alongtrack Scanning Radiometer to changing aerosol optical thickness did not agree as a function of different meteorological states. Diamond et al. (2020) demonstrate that at least 5 years of observational data are needed to detect sufficient signal-to-noise in cloud properties to a change in aerosol (where aerosol sources are known, in this case a shipping corridor). Thus, this study may be at the limits of having enough observational data collected to be a useful constraint for aerosol-cloud interactions after sub-division of the data into meteorological composites. Although, these smaller but detailed ground-based data sets are essential for model evaluation when the methods applied to nudged models by reanalysis data and observations are consistent.

One targeted improvement to the observations might be to incorporate ground-based retrievals of precipitation using a scanning radar. This would improve detection of drizzle along trajectories; however, this type of data is sparse in the region and the horizontal range of a typical X-band radar cannot cover a typical multi-day trajectory which spans hundreds to thousands of kilometers. Complete coverage via other ocean-based platforms or spaceborne retrievals of precipitation at geostationary orbit are not feasible or possible in the near-term future. Combining the retrievals from CloudSat with those from other passive radars has shown useful for increasing the detection of light precipitation (Eastman et al., 2019). In addition, satellite retrieved $R_e$ and cloud optical thickness are used to estimate $N_d$ have large uncertainties from passive remote sensors (upwards of 80% for Grosvenor et al., 2018) due to invoking a variety of assumptions on the cloud state (one being cloud adiabaticity Merk et al., 2016) in the retrieval calculation and instrument limitations described in (Grosvenor et al., 2018) and may be better constrained when combined with detailed ground-based ARM retrievals.

Finally, a series of autoconversion and accretion experiments were carried out to determine whether the sensitivity of the warm rain process is tied to the microphysical and macrophysical properties of the clouds. In general, we find only small changes in the Twomey effect amongst experiments but larger impacts on the LWP and CF radiative forcing adjustments (with an unexpected positive sign). While the radiative forcing estimates are in fair agreement with the satellite observations, E3SM



simulations suggest that aerosol activation, warm rain processes, and or turbulence parameterizations may require further modification (such as including improved parameterizations of entrainment and wet deposition) to achieve better agreement with base-state variables (such as cloud droplet concentration), and $ERF_{aci}$ compared to satellite and ARM observations.

*Code and data availability.* All ARM products listed in Table 1 are available at https://www.arm.gov/data/. CERES SYN Ed4a 4 product is available at https://ceres.larc.nasa.gov. MODIS collection 6 MYD08 D3 product is available at https://earthdata.nasa.gov. IMERG data are available from the NASA Goddard Space Flight Center; https://pmm.nasa.gov. MERRA2 data were obtained from https://goldsmr4. gesdisc.eosdis.nasa.gov/data/MERRA2/. HYSPLIT trajectory code is available at https://www.ready.noaa.gov/HYSPLIT.php. All data and code availability websites were last accessed on 17 August 2022.

*Video supplement.* Movies S1 and S2 related to this article are available in the supplementary materials.

*Author contributions.* MWC wrote the manuscript and developed the Lagrangian trajectory approach and analysis. PLM guided the implementation of the E3SMv1 simulations. Research and development ideas as well as writing and editing were contributed by coauthors PLM, PW, AV, JM, and JF

*Acknowledgements.* This research has been supported by the "Enabling Aerosol cloud interactions at GLobal convection-permitting scalES (EAGLES)" project (74358), funded by the US Department of Energy, Office of Science, Office of Biological and Environmental Research, Earth System Model Development program. The Pacific Northwest National Laboratory is operated for the US Department of Energy by Battelle Memorial Institute under contract DE-AC05-76RL01830.



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



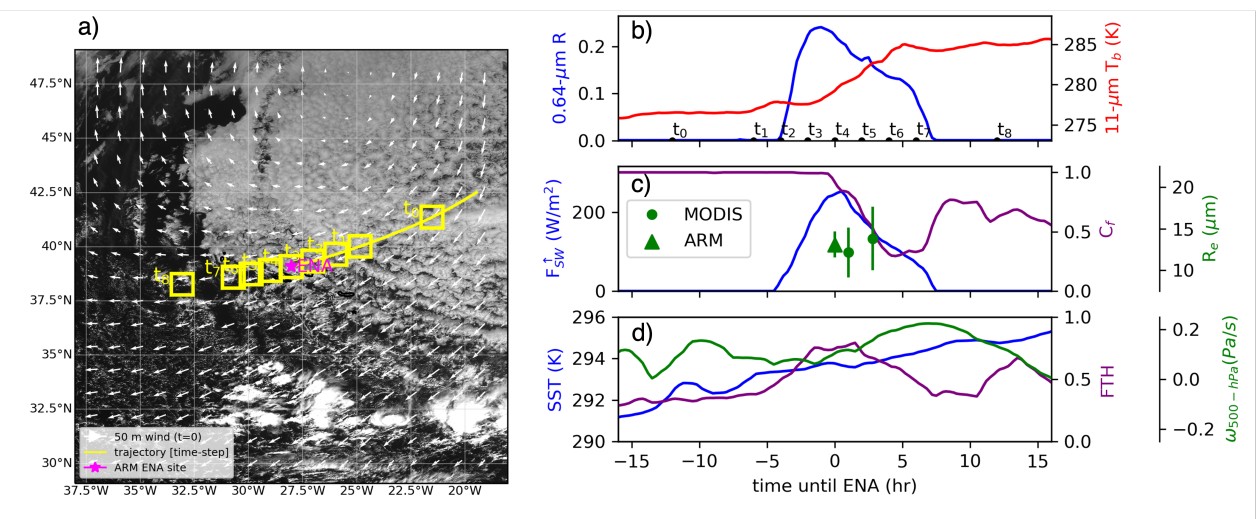

**Figure 1.** (a) Lagrangian trajectory (yellow line) initialized at ENA at 10/27/18 at 12:00 UTC computed using HYSPLIT and MERRA2 products is plotted over the GOES-16 visible image (0.64-$\mu$m reflectance). The trajectory spans two 16-hour periods (backward and forward). (b) Visible reflectance and 11-$\mu$m brightness temperature from GOES-16, (c) CERES SYN top of atmosphere (TOA) outgoing shortwave flux ($F_{SW}^{\uparrow}$, blue) and cloud fraction (purple) with droplet effective radius from MODIS (green circle) and ARM (triangle; vertical lines denote 1 standard deviation in the retrievals), and (d) sea surface temperature (SST; blue), free troposphere relative humidity at 850 hPa (FTH; purple), and subsidence rate at 500 hPa ($\omega_{500-hPa}$; green) are interpolated in time and space to the trajectory in 15 minute intervals.





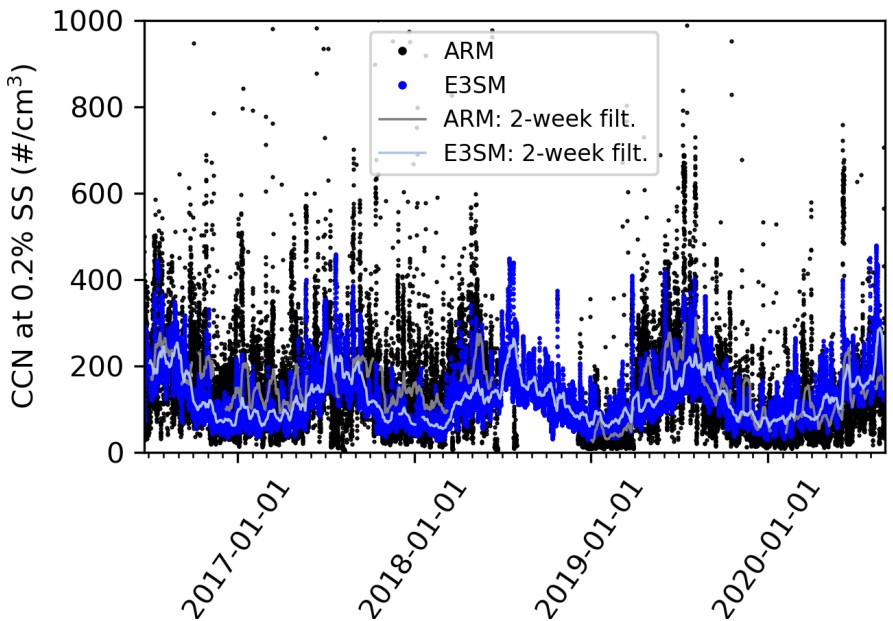

**Figure 2.** (a) Cloud condensation nuclei concentration at 0.2% super saturation are measured from ARM (black) and simulated using E3SM (blue) at the ENA site over the period 06/21/2016 – 08/29/2020. Two-week filter is applied to the hourly data for ARM (gray) and E3SM (light blue) data sets.

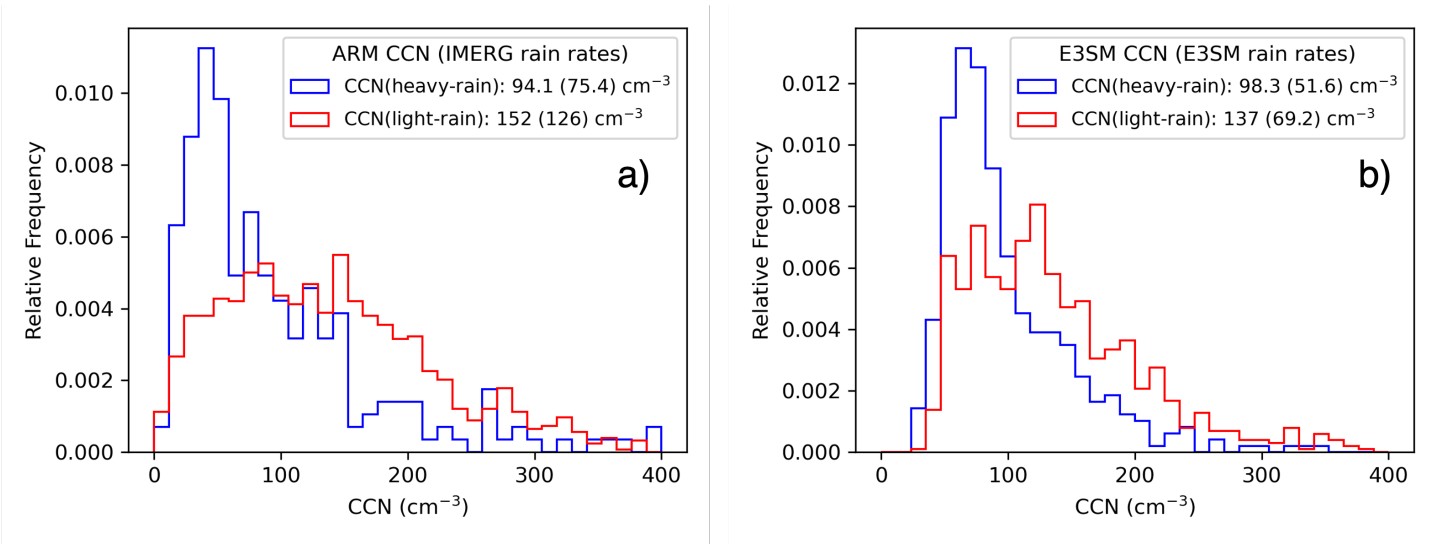

**Figure 3.** (a) Relative frequency in the distribution of CCN concentration measured at 0.2% super saturation observed by ARM at the ENA site are provided for trajectories with above (blue) and below (red) median accumulated IMERG precipitation in the previous 48-hours along the back trajectories. (b) Shows the same distributions from the same trajectories but using E3SM CCN concentration and precipitation rates instead. Means and standard deviations are provided for each distribution.





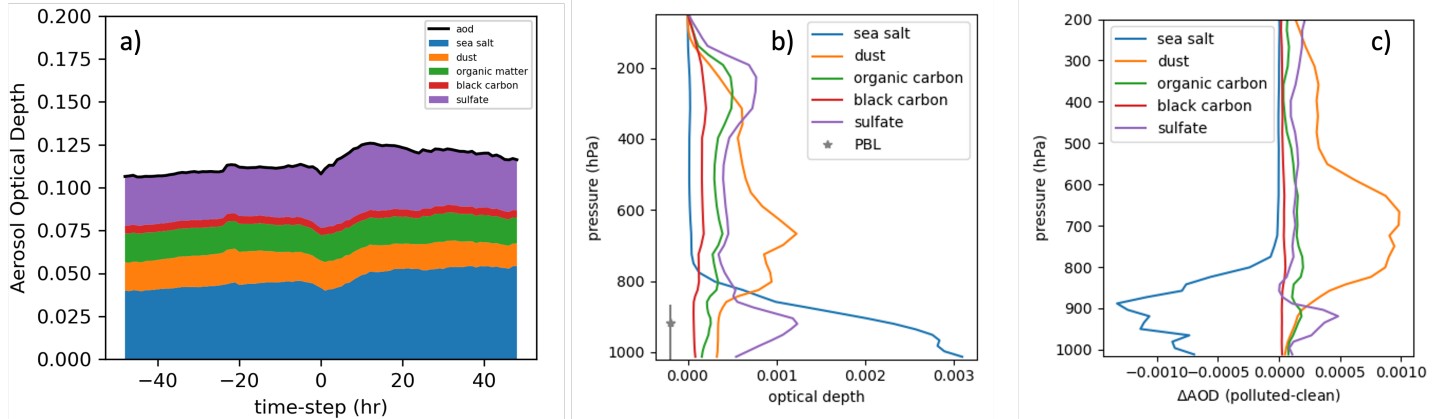

**Figure 4.** (a) Aerosol optical depth decomposed into sea salt (blue), dust (orange), organic matter (green), black carbon (red), and sulfate (purple) from MERRA2 averaged along backward and forward trajectories over the period 06/21/2016 – 08/29/2020. (b) Vertical profile of the mean aerosol optical thickness for each species in each of the 72 pressure levels at the location of the ARM site at ENA. The average PBL height plotted over two standard deviations is provided along the y-axis. (c) Change in AOD between polluted and clean trajectories based on above and below median CCN measured from ARM.


**Figure 5.** (a) CCN concentration (red) and rain rate from ARM hourly measurements. (b) CCN concentration measured at ARM averaged over the period 06/21/2016 – 08/29/2020 grouped by the amount of time until a rain event occurs with a threshold above 0.05 mm/hr. (c) The same as (a) except using simulations from the E3SM model A1R0 model run. (d) The same as (b) except using simulations from each E3SM experiment.



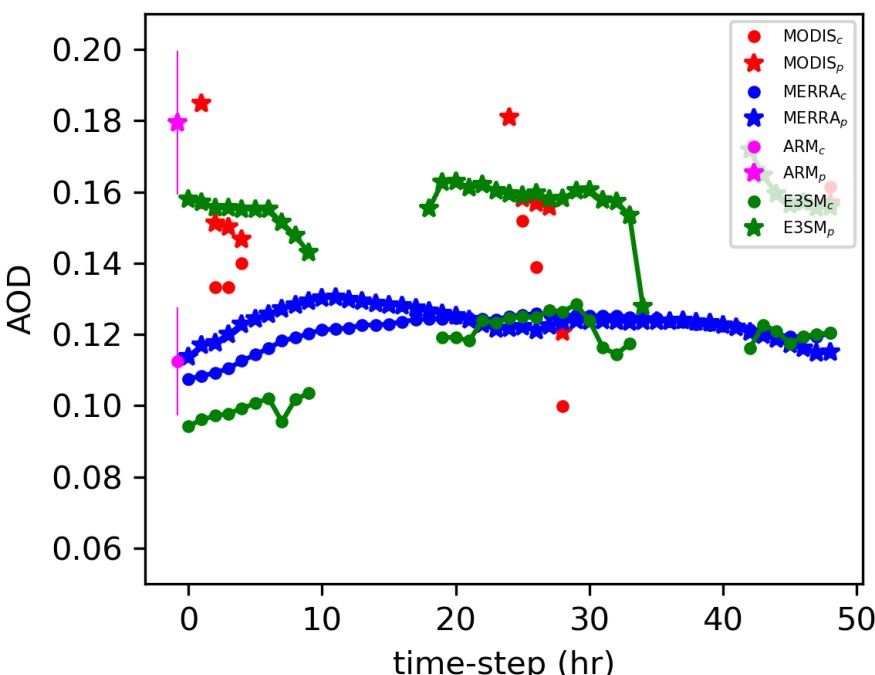

**Figure 6.** Aerosol optical depth averaged along forward trajectories in MERRA, E3SM, MODIS and ARM observations for trajectories with below (filled circle) and above (star) median initial CCN concentrations of 110 for MODIS, ARM, MERRA, and E3SM data sets.

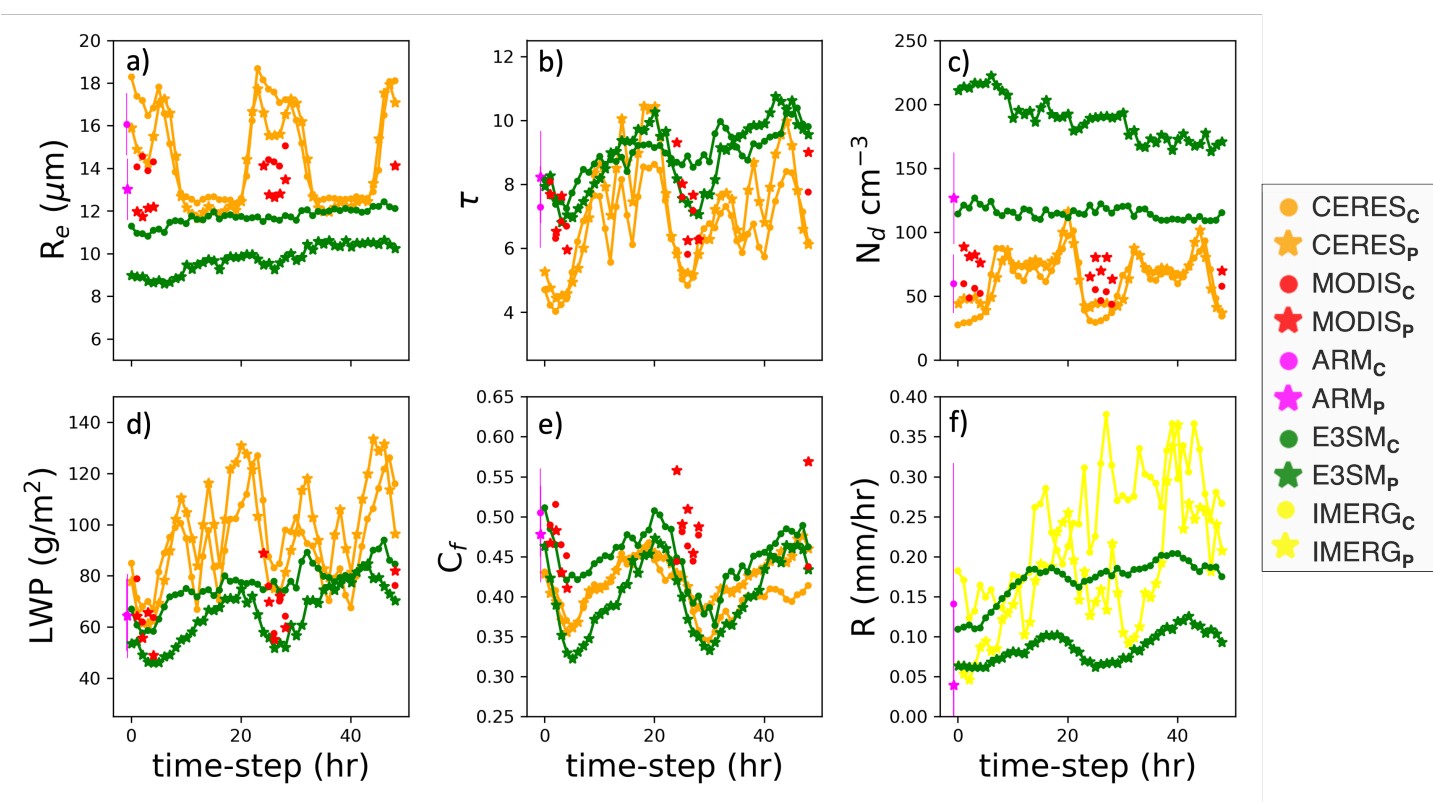

**Figure 7.** Droplet effective radius (a), cloud optical thickness (b), liquid water path (c), droplet number concentration (d), liquid cloud fraction (e), and precipitation rate (f) averaged along the forward trajectories initialized from the ENA site where the airmass is considered polluted (*) and clean (o) for CERES (orange), ARM (magenta), MODIS (red), and E3SM (green).





**Figure 8.** Difference in cloud droplet number concentration ($\Delta N_d$) between polluted (CCN > 110 cm$^{-3}$) and clean (CCN < 110 cm$^{-3}$) trajectories within 6 hours of the CCN observation for (a) ARM, (b) MODIS, and (c) E3SM. Data are stratified by below (blue) and above (red) median lower tropospheric stability (LTS), free-tropospheric humidity (FTH), vertical velocity at 500-hPa ($\omega\_500$), rain rate (non-raining and raining), coupling strength (coupled, non-coupled), amount of aerosol in the PBL (aod_pbl) and by Froude number (values 0–1 and 1–2). Error bars represent the 95% confidence interval computed from a two-tailed t-test. Hatching denotes statistically significant differences





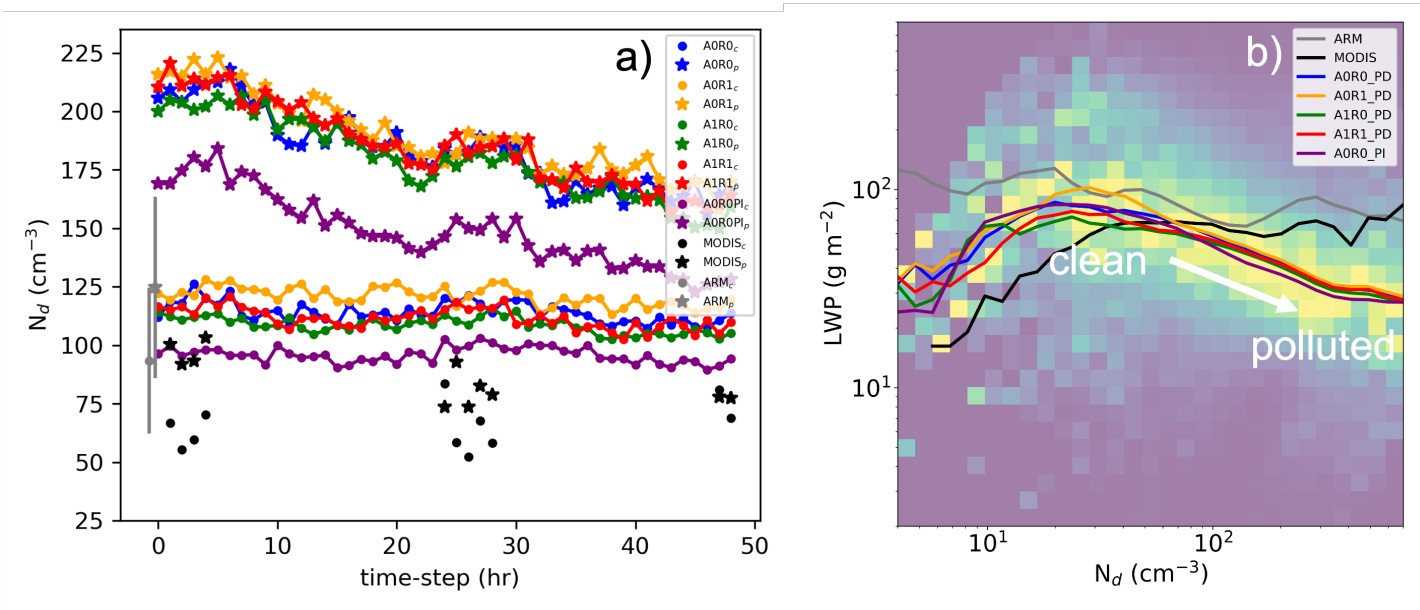

**Figure 9.** (a) Cloud droplet concentration for clean (circle) and polluted (asterisks) airmasses averaged along forward trajectories from each E3SM experiment. (b) Joint histogram of the liquid water path (LWP) and cloud droplet number concentration for E3SM. Median LWP is plotted over the joint histogram for each observational and simulation data set.





**Table 1.** Summary of ARM data products analyzed for the period of 2016-06-22 through 2020-10-28.

| Geophysical quantity | file variable name | product | time period | sampling rate |
|---|---|---|---|---|
| Cloud condensation nuclei | N_CCN | aosccn1colspectra | 2016-06-22 - 2020-10-28 | hourly |
| Cloud fraction (total sky) | tot_cld_tsi | armbecldrad | 2014-01-01 - 2020-12-31 | hourly |
| Cloud fraction (low-level liquid clouds) | cloud_layer_top_height | arsclkazr1kollias | 2015-07-17 - 2022-04-30 | 1 min |
| Rain rate | rain_rate | ldquantsC1.c1. | 2014-02-27 – 2022-08-30 | 1 min |
| Surface meteorology | temperature/relative_humidity | armbeatmC1.c1. | 2014-01-01 – 2020-12-31 | hourly |
| Cloud effective radius | effective_radius_average | mfrsrcldod1minC1.c1. | 2014-01-01 – 2020-12-31 | 20 mins |
| Cloud optical depth | optical_depth_average | mfrsrcldod1minC1.c1. | 2014-01-01 – 2020-12-31 | 20 mins |
| Liquid water path | lwp | mfrsrcldod1minC1.c1. | 2014-01-01 – 2020-12-31 | 20 mins |
| Aerosol optical thickness | aerosol_optical_depth_filter2 | mfrsraod1michC1 | 2016-01-01 – 2020-01-25 | 20 mins |
| Cloud base height | first_cbh | ceil.b1 | 2013-09-29 – 2022-02-12 | 10 mins |

**Table 2.** Names of each E3SM simulation experiment based on different parameterizations of the KK2000 autoconversion and accretion enhancement factor schemes. The final experiment represents pre-industrial (PI) based aerosol emissions using the A0R0 set up.

| Name | $a$ | $b$ | $c$ | $accre$ |
|---|---|---|---|---|
| A0R0 | 3.19 | -1.2 | 30500 | 1.5 |
| A0R1 | 3.19 | -1.2 | 30500 | 1.0 |
| A1R0 | 2.47 | -1.79 | 1350 | 1.5 |
| A1R1 | 2.47 | -1.79 | 1350 | 1.0 |
| A0R0PI | 3.19 | -1.2 | 30500 | 1.5 |





**Table 3.** Fractional change in cloud condensation nuclei (CCN) concentration at 0.2% supersaturation based on trajectories sorted by above and below median precipitation rate (with threshold 0.05 mm/hr) for the IMERG and E3SM simulation experiments listed in Table 2. Precipitation is aggregated in both the Eulerian (EUL) and Lagrangian (LAG) frames of reference.

| Experiment | $\Delta\text{CCN}_{EUL}/\text{CCN}$ | $\Delta\text{CCN}_{LAG}/\text{CCN}$ |
|---|---|---|
| IMERG | $-0.38 \pm -0.50$ | $-0.38 \pm 0.51$ |
| A0R0 | $-0.20 \pm 0.34$ | $-0.28 \pm 0.32$ |
| A0R1 | $-0.19 \pm 0.34$ | $-0.30 \pm 0.32$ |
| A1R0 | $-0.23 \pm 0.34$ | $-0.31 \pm 0.32$ |
| A1R1 | $-0.21 \pm 0.34$ | $-0.30 \pm 0.32$ |
| A0R0PI | $-0.16 \pm 0.35$ | $-0.25 \pm 0.32$ |

**Table 4.** List of radiative effects decomposed into contributions from the Twomey effect and liquid water path and cloud fraction adjustments averaged over the first day of each trajectory.

| | twomey (W m$^{-2}$) | LWP$_{adj.}$ (W m$^{-2}$) | CF$_{adj.}$ (W m$^{-2}$) | net effect (W m$^{-2}$) |
|---|---|---|---|---|
| MODIS | -1.16±0.13 | 0.21±0.02 | 0.49±0.05 | -0.46±-0.09 |
| CERES | -0.72±0.34 | 0.55±0.26 | 0.08±0.04 | -0.10±-0.08 |
| ARM | -1.82±0.44 | 0.56±0.13 | 1.24±0.30 | -0.03±-0.01 |
| A0R0 | -1.56±0.20 | 1.46±0.19 | 1.21±0.16 | 1.11±0.25 |
| A0R1 | -1.59±0.19 | 1.32±0.16 | 1.55±0.19 | 1.27±0.27 |
| A1R0 | -1.61±0.18 | 1.04±0.12 | 1.51±0.17 | 0.95±0.18 |
| A1R1 | -1.60±0.19 | 0.98±0.11 | 1.54±0.18 | 0.92±0.18 |
| A0R0PI | -1.56±0.21 | 1.09±0.15 | 1.12±0.15 | 0.66±0.16 |

**Table 5.** List of quantities used to compute the aerosol indirect radiative effect averaged over the first day of each trajectory.

| | $\overline{f_c}$ | $\overline{\alpha_c}$ | $\overline{N_d}$ (cm$^{-3}$) | $\Delta \ln N_d$ | $\Delta \ln LWP$ | $\Delta \ln CF$ |
|---|---|---|---|---|---|---|
| MODIS | 0.47±0.03 | 0.30±0.01 | 72.1±2.69 | 0.40±0.03 | -0.03±0.10 | -0.08±0.26 |
| CERES | 0.39±0.03 | 0.23±0.01 | 39.5±1.42 | 0.35±0.16 | -0.11±0.03 | -0.03±0.31 |
| ARM | 0.47±0.01 | 0.50±0.01 | 109±9.65 | 0.55±0.16 | -0.07±0.10 | -0.09±0.09 |
| A0R0 | 0.42±0.04 | 0.33±0.010 | 167±2.58 | 0.58±0.02 | -0.22±0.06 | -0.18±0.44 |
| A0R1 | 0.42±0.04 | 0.34±0.009 | 174±1.88 | 0.58±0.03 | -0.19±0.05 | -0.22±0.38 |
| A1R0 | 0.41±0.04 | 0.32±0.01 | 161±1.13 | 0.61±0.01 | -0.16±0.04 | -0.25±0.40 |
| A1R1 | 0.41±0.04 | 0.32±0.01 | 167±1.68 | 0.61±0.03 | -0.15±0.07 | -0.25±0.37 |
| A0R0PI | 0.42±0.04 | 0.31±0.009 | 139±2.79 | 0.59±0.03 | -0.17±0.05 | -0.19±0.37 |





**Table 6.** List of radiative effects decomposed into contributions from the Twomey effect and liquid water path and cloud fraction adjustments averaged over the second day of each trajectory.

|  | twomey (W m$^{-2}$) | LWP$_{adj.}$ (W m$^{-2}$) | CF$_{adj.}$ (W m$^{-2}$) | net effect (W m$^{-2}$) |
|---|---|---|---|---|
| MODIS | -0.98±0.29 | -0.20±0.06 | -0.71±0.21 | -1.88±-0.99 |
| CERES | -0.13±0.62 | 0.29±1.38 | -0.11±0.52 | 0.05±0.40 |
| A0R0 | -1.17±0.13 | 2.01±0.23 | 0.66±0.07 | 1.51±0.29 |
| A0R1 | -1.14±0.13 | 2.06±0.24 | 1.17±0.13 | 2.09±0.42 |
| A1R0 | -1.24±0.10 | 1.37±0.11 | 1.05±0.09 | 1.17±0.17 |
| A1R1 | -1.29±0.10 | 1.75±0.13 | 1.07±0.08 | 1.53±0.20 |
| A0R0PI | -1.01±0.11 | 2.02±0.22 | 0.48±0.05 | 1.48±0.28 |

**Table 7.** List of quantities used to compute the aerosol indirect radiative effect averaged over the second day of each trajectory.

|  | $\overline{f_c}$ | $\overline{\alpha_c}$ | $\overline{N_d}$ (cm$^{-3}$) | $\Delta \ln N_d$ | $\Delta \ln LWP$ | $\Delta \ln CF$ |
|---|---|---|---|---|---|---|
| MODIS | 0.50±0.03 | 0.31±0.02 | 63.9±5.42 | 0.30±0.06 | 0.02±0.08 | 0.09±0.19 |
| CERES | 0.39±0.02 | 0.27±0.01 | 45.1±10.1 | 0.10±0.26 | -0.09±0.11 | 0.05±0.30 |
| A0R0 | 0.38±0.03 | 0.34±0.008 | 153±2.60 | 0.46±0.04 | -0.32±0.05 | -0.10±0.35 |
| A0R1 | 0.39±0.03 | 0.35±0.009 | 155±2.77 | 0.43±0.03 | -0.31±0.05 | -0.17±0.32 |
| A1R0 | 0.38±0.03 | 0.32±0.006 | 147±2.80 | 0.50±0.02 | -0.22±0.05 | -0.18±0.32 |
| A1R1 | 0.39±0.02 | 0.33±0.008 | 152±2.73 | 0.51±0.02 | -0.28±0.04 | -0.17±0.29 |
| A0R0PI | 0.39±0.03 | 0.32±0.007 | 126±2.64 | 0.40±0.03 | -0.32±0.07 | -0.08±0.31 |

**Table 8.** List of cloud and radiative effects from aerosol perturbations based on present-day relative to pre-industrial type simulations.

|  | present day | pre-industrial | $\Delta(PD - PI)$ |
|---|---|---|---|
| $R_e$ [$\mu$m] | 10.00±2.69 | 10.7±2.81 | -0.68±0.05 |
| $\tau_c$ | 9.71±6.15 | 8.94±5.49 | 0.77±0.09 |
| $LWP$ [g/m$^2$] | 69.7±56.7 | 68.4±54.2 | 1.27±0.77 |
| $N_d$ [#/cm$^3$] | 154±83.7 | 125±67.1 | 29.1±2.08 |
| $CF$ | 0.42±0.34 | 0.42±0.34 | 0.002±0.003 |
| $F_{SW}^{\uparrow}$ [W/m$^2$] | 77.6±23.2 | 75.25±22.3 | 2.41±0.44 |
| Twomey [W/m$^2$] |  |  | -1.56±0.39 |
| $LWP_{adj.}$ [W/m$^2$] |  |  | -0.31±0.08 |
| $CF_{adj.}$ [W/m$^2$] |  |  | -0.16±0.04 |
| $ERF_{aci}$ [W/m$^2$] |  |  | -2.04±-0.22 |