# Peer review of "Evaluation of Aerosol-Cloud Interactions in E3SM using a Lagrangian Framework"

_Atmospheric Chemistry and Physics, 2022_

## Referee Comment (RC1)

In this study, the authors applied a Lagrangian framework to the E3SM to evaluate the aerosol-cloud interactions against multiple observations and reanalysis. Additionally, the framework uses direct measurements of CCN instead of AOD, and sensitivity tests to the parameters in the autoconversion and accretion scheme were performed. The biases in the E3SM simulation and possible causes were discussed. Given that the Lagrangian framework can provide some additional information for model evaluation compared to the common Eulerian perspective and the manuscript overall is well structured, I think this manuscript is suitable for the publication in ACP after addressing the following concerns.

**General Comments:**

In Section 4.1, specifically in Figure 5, the authors investigated the relationship between CCN and rain rate. For different E3SM sensitivity tests, there are no obvious differences among different present-day simulations. Since aerosol wet removal can be done both in and below clouds and model can diagnose these processes, I suggest that the authors show the differences of in-cloud and below-cloud wet scavenging among different simulations in detail. It might also be useful to understand why $\frac{\Delta CCN}{\Delta R}$ in the ARM observations is larger than that in the E3SM simulations.

In Figure S7, for a given rain rate, ARM features higher efficiency in washing out aerosols than the model, especially for rain rates smaller than $10^{-3}$ mm/hr. Why?

**Specific Comments:**

L187: Please define $N_d$ in the equation.

L233: A brief description of the Froude number and the threshold indicating the flow was not blocked is expected.

Figure 2: ARM shows larger variations than E3SM. Why?

Figure S4: I did not find that means and standard deviations are provided.

Figure 3: Are the values in the parenthesis standard deviations?

L265-L273: In this study, the authors define light rain smaller than the accumulated average threshold of 0.05 mm/hr and heavy rain larger than this threshold. However, according to previous studies (e.g., Chen et al., 2021; Wang et al, 2021), light rain is often defined as a daily average smaller than 10 or 20 mm/day. Following these criteria, heavy rain defined here should also be grouped into the light rain category. Therefore, I suggest that the authors change the terminology of light and heavy rain to rain below and above the accumulated average threshold of 0.05 mm/hr, respectively throughout

the text.

Section 4.5.2: For the first approach, the difference terms represent differences in cloud properties between the clean and polluted state based on measured values of the CCN at the ARM site. However, for $\frac{\tau_a^{PD} - \tau_a^{PI}}{\Delta \tau_a}$ in Eq. (5), how to derive it in observations? Is the term equal to 1? If yes, please clarify this.

For the second approach, did the derived ERFaci significantly differ from the value using the method stated in Ghan (2013)? In model simulations, ERFaci can be calculated as $\Delta(F_{clean} - F_{clean,clear})$, where $\Delta$ denotes the difference between PD and PI, and each term can be directly output by the model.

**References:**
Chen, D., Dai, A., & Hall, A. (2021), The convective-to-total precipitation ratio and the "drizzling" bias in climate models. *Journal of Geophysical Research: Atmospheres*, 126, e2020JD034198. https://doi.org/10.1029/2020JD034198.

Wang, Y., G. J. Zhang, S. Xie, W. Lin, G. C. Craig, Q. Tang, and H.-Y. Ma (2021), Effects of coupling a stochastic convective parameterization with the Zhang–McFarlane scheme on precipitation simulation in the DOE E3SMv1. 0 atmosphere model, *Geoscientific Model Development*, *14*(3), 1575-1593.

---

## Author Comment (AC1)

**Evaluation of Aerosol-Cloud Interactions in E3SM using a Lagrangian Framework**

**Referee Comments**

Point-by-point responses in blue, additions to manuscript are ***bold & italicized***
References used to reply to comments are listed at the end.

Dear Referees,

Thank you for your time and effort reviewing our manuscript. We appreciate that you provided a fair and insightful evaluation. Your comments have led to key changes in the manuscript that, we believe, improved the clarity and accuracy of the analysis. Specifically, the changes include an infographic which better depicts the integrated data sets, a complementary analysis of $ERF_{aci}$ using the Ghan (2013) method, and an investigation into the impacts of in-cloud and below-cloud wet scavenging on the aerosol size distributions in E3SM. Overall, the narrative is similar, and with these clarifications and modifications we believe the conclusions are stronger in the revised manuscript.

Best regards,
Matt
* * *
**Anonymous Referee #1**

In this study, the authors applied a Lagrangian framework to the E3SM to evaluate the aerosol-cloud interactions against multiple observations and reanalysis. Additionally, the framework uses direct measurements of CCN instead of AOD, and sensitivity tests to the parameters in the autoconversion and accretion scheme were performed. The biases in the E3SM simulation and possible causes were discussed. Given that the Lagrangian framework can provide some additional information for model evaluation compared to the common Eulerian perspective and the manuscript overall is well structured, I think this manuscript is suitable for the publication in ACP after addressing the following concerns.

**General Comments:**
In Section 4.1, specifically in Figure 5, the authors investigated the relationship between CCN and rain rate. For different E3SM sensitivity tests, there are no obvious differences among different present-day simulations. Since aerosol wet removal can be done both in and below clouds and model can diagnose these processes, I suggest that the authors show the differences of in-cloud and below-cloud wet scavenging among different simulations in detail.
>> Thank you for raising this key point. We have examined the wet removal of both in- and below-cloud wet scavenging in E3SMv1 for each of the sensitivity tests described in this study. Figure R1 (below; and as **Figure S8** in the revised manuscript) shows the mean values from each autoconversion and accretion experiment for the raining and non-raining trajectories normalized by total column burden aerosol concentration for each mode. As expected, significantly larger sinks of the accumulation and Aitken mode (as well as the coarse mode, not shown) aerosols occur in trajectories with heavier precipitation in all sensitivity experiments (gray rectangles). There are also notable differences between autoconversion experiments

which are more apparent in Table R1 (also provided as **Table S1** in the revised manuscript). Generally, the below-cloud scavenging sink is smaller than the in-cloud scavenging sink (Fig R1) and most of the difference between experiments is through in-cloud scavenging. The A1 based experiments (with more sensitivity to droplet number concentrations) are approximately 15 - 30% larger than the A0 based experiments but despite this stronger autoconversion rate the CCN concentrations remain roughly similar which may imply other processes are at play (described later in later comments) that may be important but outside of the scope of the current study. We have added the following plot and table to the manuscript since we believe that they add value and greater understanding of the model system behavior.

[Figure]

Figure R1, Mean wet scavenging rates for below- (a, and c) and in-cloud (b and d) accumulation-mode aerosols separated by trajectories with below (black; non-raining) and above (gray; raining) median precipitation rates at Graciosa Island shown for each sensitivity experiment (A0R0, A0R1, A1R1, A1R1, PI).

Table R1, Difference in below-cloud and in-cloud wet scavenging rates between different autoconversion experiments (A0R0, A1R0). Bolden font indicates that the difference is statistically significant at the 95th percentile.

| Wet scavenging rate (relative change between experiments) | (A1R0 – A0R0)/A0R0 | (A1R1 – A0R1)/A0R1 |
| --- | --- | --- |
| Below-cloud accumulation mode | -0.03 | -0.04 |
| Below-cloud aitken mode | 0.07 | -0.03 |
| Below-cloud coarse mode | -0.03 | -0.015 |
| In-cloud accumulation mode | **0.15** | **0.14** |
| In-cloud aitken mode | **0.29** | **0.29** |
| In-cloud coarse mode | **0.22** | **0.23** |

It might also be useful to understand why $\frac{\Delta CCN}{\Delta R}$ in the ARM observations is larger than that in the E3SM simulations. In Figure S7, for a given rain rate, ARM features higher efficiency in washing out aerosols than the model, especially for rain rates smaller than 10-3 mm/hr. Why?

>> The sudden decline in the ARM observations around $10^{-3}$ is likely due to this bin containing more non-precipitating clouds than that simulated in E3SM. Nevertheless, the overall slope over the whole range in rain rates is more negative in the observations compared to the model. We have added the following to the caption of FigS7:

> ***Note, the first bin can include non-precipitating clouds.***

**Specific Comments:**

L187: Please define Nd in the equation.
>> Nd is the cloud droplet number concentration which is now defined here.

L233: A brief description of the Froude number and the threshold indicating the flow was not blocked is expected.
>> We have added a brief description of the Froude number as described in Ebmeier et al. (2014) and the differences in cloud patterns between blocked (leewave wakes) vs unblocked (vortex shedding) downstream flows. We have also modified the bullet point in the conclusions to read as follows:

> ***Although, an island wake effect can be observed in individual case studies (e.g. Figure 1), analysis of the Froude number suggests leeward wakes capable of strongly influencing cloud properties is small on average (Figures S14 and S15).***

Figure 2: ARM shows larger variations than E3SM. Why?
>> The black and blue dots represent hourly timescales and larger variation in CCN concentration are shown in the ARM measurements compared to the E3SM simulations, this is also depicted in Figures 4a and b where larger standard deviations are shown in the ARM CCN data compared to E3SM. We have included this point in the manuscript by adding the following sentences:

> ***On average, ARM shows larger variations in hourly measurements of CCN concentration compared to the E3SM model but these variations tend to decrease at longer averaging time-scales (e.g. 2-week running mean shows similar characteristics). The larger hourly variability may be caused by differences in the following: 1) spatial scales between point-location and grid-box mean values as suggested by Schutgens et al. (2016) land area representation in E3SM, and 3) local-scale island emissions.***

Figure S4: I did not find that means and standard deviations are provided.
>> This was an plot importing error. Figure captions S4 and S5 were in switched order and have now been corrected.

Figure 3: Are the values in the parenthesis standard deviations?
>> Yes, I have added "…standard deviations **as shown in parenthesis** are provided…." to the figure caption. This was also corrected in Figure S5.

L265-L273: In this study, the authors define light rain smaller than the accumulated average threshold of 0.05 mm/hr and heavy rain larger than this threshold. However, according to previous studies (e.g., Chen et al., 2021; Wang et al, 2021), light rain is often defined as a daily average smaller than 10 or 20 mm/day. Following these criteria, heavy rain defined here should also be grouped into the light rain category. Therefore, I suggest that the authors change the terminology of light and heavy rain to rain below and above the accumulated average threshold of 0.05 mm/hr, respectively throughout the text.
>> Thank you for your suggestion. We agree, the terminology "light" and "heavy" is ambiguous which could lead to differing interpretations on the CCN concentration response as it relates to

other studies. Therefore, we have changed the text to the following: "Trajectories are sorted into relatively low and high values of the accumulated average precipitation rate with a threshold of 0.05 mm/hr (depicted in Figure S4)."

Section 4.5.2: For the first approach, the difference terms represent differences in cloud properties between the clean and polluted state based on measured values of the CCN at the ARM site. However, for $\frac{\tau_a^{PD}-\tau_a^{PI}}{\Delta\tau_a}$ in Eq. (5), how to derive it in observations? Is the term equal to 1? If yes, please clarify this. .

>> This term represents the anthropogenic aerosol fraction as derived from aerosol optical depth based on the present-day and pre-industrial based aerosol emission simulations. We have clarified in the text that this factor is computed solely from the E3SM model since $\tau_a^{PI}$ cannot be obtained using satellite observations.
See inserted text:

*Because pre-industrial aerosols cannot be obtained from satellite observations, we use E3SM to represent the anthropogenic aerosol fraction ( $\frac{\tau_a^{PD}-\tau_a^{PI}}{\Delta\tau_a}$ ) which has a mean value of 0.392 for ERF$_{aci}$ estimates in both observations and models.*

Please refer to the PDF below for the changes in aerosol optical thickness between the PD and PI simulations. As expected, the difference is positive.

[Figure]

*Figure R2: Histogram of the change in aerosol optical thickness (AOD) between present-day and pre-industrial based aerosol E3SM simulations at Graciosa island. The mean and standard deviation (shown in parenthesis) is provided in the plot.*

For the second approach, did the derived ERFaci significantly differ from the value using the method stated in Ghan (2013)? In model simulations, ERFaci can be calculated as $\Delta(F_{clean} - F_{clean,clear})$, where $\Delta$ denotes the difference between PD and PI, and each term can be directly output by the model.

>> Thank you for suggesting an alternate approach to compute ERFaci. We re-ran the E3SM simulations to include net radiative flux variables at the top of the model assuming clearsky and

clean-clearsky conditions to compute ERFaci using the Ghan (2013) approach. Overall, the estimates are nearly identical. We think that this complementary analysis strengthens our findings so we have included the result in the manuscript. See inserted text:

*As a final check, we have estimated ERFaci using the same approach as that described in (Ghan, 2013). This method uses the difference in cloud radiative forcing between present-day and pre-industrial based aerosol emissions, i.e. $\Delta C_{clean} = \Delta(F_{clean} - F_{clear,clean})$, where $F_{clean}$ is the top of model net radiative flux neglecting the scattering and absorption of solar radiation by all of the aerosol and $F_{clear}$ is the flux calculated as a diagnistoc with clouds neglected. Using this method we estimate $ERF_{aci}$ to be −2.011 W/m$^{-2}$ which is nearly identical in strength to the combined Twomey, LWP$_{adj}$, and CF$_{adj}$ radiative effects discussed previously (Table 8).*

**Anonymous Referee #2**

The proposed paper evaluates the E3SM model skill at representing aerosol-cloud interactions (ACI) in warm rain clouds. The E3SM model is compared against the satellite as well as the ground based measurements from the ARM site in the Azores using a Lagrangian framework. The paper is well written and skillfully uses many different observations and an impressive number of trajectories to evaluate the E3SM model. I suggest that the paper should be published after some minor revisions.

**Main comments:**
I think that the amount and synergy of the observational data used to evaluate the E3SM model is really impressive, and it would be great to highlight it even more. Would it, for example, be possible to add an infographic summarizing all the lines of evidence used in this work? If not, then maybe a table summarizing all the used data could be included in the main text?
>> We with your suggestion and added a new figure (Figure R3, also now *Figure 1*) to the manuscript which includes an infographic of the datasets used. We also have added details for the satellite data described in Table 1.

[Figure]

*Figure R3: Information graphic showing the satellite (orange dashed arrow) and surface ARM (red dashed arrow) data sets in trajectories passing over Graciosa Island.*

It would be helpful to the reader to explain in more detail how the ACI are represented in the E3SM model. The references to the papers describing different parameterizations are provided, but the description of how the coupling between them works is missing. That would be useful to understand, especially when later trying to reason about the E3SM results for different aerosol scenarios and the lack of the desired connection between aerosols and precipitation.
>> We agree and have added more detail to better describe the coupling between the aerosols, cloud, and precipitation schemes so that ACI can be more easily interpreted from our results. Note, that because the manuscript is already quite long we could not incorporate all of this detailed information. Please see detailed answers to your questions below.

Could the following be described:
- How is the microphysics scheme coupled to CLUBB, deep convection parameterization and the large scale flow?

>> EAMv1 uses the version 2 of Morrison and Gettelman (MG2) two-moment bulk microphsyics scheme (Gettelman and Morrison, 2015) and is coupled with CLUBB and MAM4. As described in Rasch et al., (2019), CLUBB prognoses eight prognostic variables (representing subgrid higher-order moments of turbulence as well as the mass and number of small and large liquid and ice particles). The Abdul-Razzak and Ghan (2000) scheme is used for aerosol activation/droplet nucleation. For ice nucleation, EAMv1 uses a classical nucleation theory (CNT)-based parameterization for the heterogeneous ice formation in mixed-phase clouds which depends on both dust and black carbon (Wang et al., 2014). The homogeneous ice nucleation of sulfate is based on Liu et al (2007). EAMv1 does not consider detailed microphysical aerosol-cloud interactions in deep convection, but includes a unified treatment for convective transport and scavenging of aerosols with secondary activation in convective updrafts above cloud base (Wang et al., 2013).

- And a clarifying question: is the KK2000 scheme part of the Gettelman and Morrison 2015 scheme?
>> The following has been clarified in the text. **KK2000 is part of MG2 but in E3SMv1 the coefficients have been changed to ensure a better fidelity of the climate simulations.** Please see Rasch et al. (2019) for details.

- How is the MAM4 aerosol model coupled to the microphysics scheme? This includes describing (i) how the available aerosol number concentration is translated into CCN and cloud droplet number concentration in the CLUBB+KK2000+MAM4 framework;
>> Aerosol activation for the multi-modal aerosol size distribution is represented based on the Abdul-Razzak and Ghan (2000) scheme. Activation occurs at cloud base as a function of temperature, pressure, updraft velocity, and aerosol chemical and physical properties. The updraft velocity is estimated from CLUBB's prognostic variable subgrid vertical velocity variance. EAMv1 also treats aerosol activation when cloud grows (i.e., when cloud fraction increases from the previous timestep). The EAMv1 autoconversion (which is a variant of KK2000) scheme is a function of cloud water mass and number mixing ratios. This scheme first computes the cloud water mass tendency, and then computes the number tendency from the mass tendency by assuming the rain droplet size of 25 um. The wet scavenging is based on Rasch et al. (2000): In-cloud scavenging depends on cloud fraction, cloud water, and precipitation production profiles for computing a first-order loss rate for cloud water. The cloud water loss rate profiles are multiplied by a solubility factor for the first-order loss rates of aerosols. Below cloud scavenging rates are computed as solubility factor x scavenging coefficient times precipitation rate. The solubility factor and scavenging coefficients are considered as tunable parameters as described in Wang et al (2013).

(ii) how rain affects the aerosol number concentration in different MAM4 modes in the CLUBB+KK2000+MAM4 framework?
>> Rain has a significant impact on both the interstitial and cloud-borne aerosol number concentration through via in- and below-cloud scavenging (See Table R1 above). This has an evident impact on the Aitken and Accumulation aerosol size distributions in MAM4 (see R5 below).

[Figure]

Figure R4 Lognormal size distribution representing MAM using the mean aerosol radius and total aerosol burden below cloud output from the model and fixed sigma values for accumulation (blue) and Aitken (red) modes for trajectories in which the median rainfall was below (dashed) and above (solid) 0.05 mm/hr for the present-day A0R0 simulation.

- For completeness could the formula for accretion rate also be provided (around page 6 line 185).
>> The following sentence has been added for completeness.

**Accretion rate is expressed in KK2000 as** $P_{acc} = \left(\frac{\partial q_r}{\partial t}\right)_{acc} = F1 * F2 * 67(Q_c Q_r)^{1.15}\rho^{-1.3}$**, where Qr is the rainwater content and ρ is the air density, F1 is the sub-grid variability in Qc predicted by CLUBB, and F2 is the micro_mg_accre_enhan_fac which varies by 50% depending on the experiment (see Table 2).**

- Could an example of the current clean, polluted and pre-industrial aerosol size distributions be plotted as seen by MAM4 model?
>> Thanks for your suggestion. We have added Figure R5 to the manuscript supplementary materials (*Figure S8*). As expected, there is a larger concentration of Aitken, accumulation, and coarse mode aerosol under present day polluted conditions compared to present-day clean and pre-industrial based aerosol simulations.

[Figure]

Figure R5 Lognormal aerosol size distribution for interstitial aerosols modeled by MAM4 using total column number, mean aerosol radius, and fixed sigma values for accumulation (blue), Aitken (red), and coarse (green) modes for present day clean (dashed), polluted (solid), and pre-industrial (dotted) conditions.

How were the values of the autoconversion and accretion parameters chosen for the sensitivity experiments? What is the (micro)physical interpretation of the different setups?
The autoconversion and accretion parameters were chosen to show the impact of the autoconversion rate on ACI. Exponent values were chosen to span a range of plausible autoconversion rates (Wood et al. 2005; details of the implantation are in Rasch et al. (2019) Table 1A). Table R1 shows the average change in autoconversion rate between E3SM simulations. On average, the A1(R0R1) experiment have 15-30% larger wet deposition rates compared to A0(R0R1). Based on this difference we expect this range to be sufficient to affect the parameter space of cloud properties. The general interpretation is that by reducing the exponent on $N_d^b$, , i.e. going from A0(b=-1.2) to A1(b=-1.79) type experiments, the aerosol effect on precipitation suppression is weakened and the change in liquid water paths is reduced. This effect is shown in Tables 5 and 7, respectively.
            **These points have been elaborated in the text.**

As shown in Figure 5 (b) and (d) and discussed in text: The ARM observations show more gradual rate of CCN decrease before t=rain. In contrast, E3SM shows some CCN buildup and then a very sharp decrease before the t=rain. After t=rain the ARM measurements show a sharp but steady CCN rate of increase. In contrast, E3SM predicts a very sharp initial CCN increase, followed by a much slower growth.

Is the CCN buildup in E3SM before the t=rain explained only by the lack of efficient wet deposition? Are there other possible issues related to the MAM4 model, and the way it is connected to KK2000 and CLUBB, that would explain this? What is the possible explanation for the differences after t=rain?

>> The sharp decrease/increase in E3SM about t=0 is not very large in an absolute change compared to the observations (i.e. the y axes for E3SM are shrunken down compared to observations). This point has been clarified in the text. Furthermore, there does not appear to be a lack of wet deposition (see earlier points) and hence the differences here may be related to other processes which we elaborate further in this revision (see inserted text in conclusions of the paper).

> ***The E3SMv1 model showed that there is less efficient CCN concentration loss by precipitation compared to ARM observations. This may be due to the lack of mesoscale cloud systems which cannot be properly simulated at coarse 1 degree scales. GCMs with higher resolution (typically 25-50 km scales) generally show improvement in the simulation precipitation frequency and intensity (Haarsma et al., 2016) unless the parameterization schemes are not suited to resolution (Xie et al., 2018). In addition, a lack of strong aerosol perturbations caused by local-scale aerosol sources (e.g. airports) may not be represented well at these coarse scales and potentially affect aerosol-cloud susceptibility. Lastly, the choice of the solubility factor and scavenging coefficient in the aerosol wet deposition scheme in EAMv1 (based on Wang et al., 2013) for improving long range transport of aerosols could contribute to the lower scavenging efficiency compared to the point measurements. In future work we plan to examine these impacts from a regionally refined mesh (Tang et al., 2019) where precipitation and highly concentrated aerosol plumes may not be overly smoothed by the coarse grid and compare better to observations.***

**Minor comments:**

Would it be possible to keep the y-axis the same between the observations and E3SM model results? (For example Figures 3, 5, 8)? Maybe providing additional zoom-in for cases where the model and observations are too far apart? I think it would make the comparisons between the model and observations easier.

>> Figure 3: y-axis was adjusted to be the same value for obs and E3SM. When the y-axis is the same in figures 5 and 8 the variations in the E3SM model are more difficult to see in these plots, therefore, we have modified the caption of figure 5 by adding the following sentence: "Note, the range in CCN (y-axis) values is larger in ARM compared to E3SM." The caption for figure 8 was also modified accordingly.

Fig 4a: The legend font is too small.

>> Thanks. The legend font has been increased to increase clarity. We have also removed the aerosol legends from panels b and c. The font was also increased Fig S11.

Fig. 6 and following: It is hard to tell the pink and red colors apart.

>> ARM (pink) color has been changed to ARM (black) color to increase clarity.

Fig. 6 and following: It is hard to tell the stars and circles apart, especially in the areas where they are clustered. Maybe some different shape or solid vs not-filled shape would be better?

>> Solid circle was changed to unfilled square. We believe this makes the contrast between clean/polluted symbols more obvious.

Fig. 7: The yellow color is hard to see. >> Changed the yellow color to gray, we also used the same symbol format as Figure 6 for better consistency between them.

First line in section 3: - Should it be 3 exceptions? >> Good catch, yes, I have changed the word "two" to "three"

Page 9 line 281: Either "are" or "may"? >> The word "are" has been deleted.

**Anonymous Referee #3**

**General comments**
This paper investigates aerosol-cloud interactions of warm clouds. A huge number of Lagrangian trajectories (> 1,500) are extracted from various datasets (ARM site observation, satellite retrievals, and earth system model simulations) to understand aerosol-cloud interactions, especially precipitation-aerosol and aerosol-cloud radiative effect relations. I think this work is interesting and scientifically valuable. Some comments are below.

**Scientific comments**
As authors already know, aerosol-cloud interactions depend on cloud characteristics. This paper provides some features about the clouds over the studied region, but those are scattered. It would be helpful if the authors could add one paragraph for some information about the clouds over this region.
>> We have added a few sentences to section 2.1 which provide more detail and context to the clouds studied here.
*Graciosa Island resides in the boundary of the subtropics and the midlatitudes and experiences meteorological conditions in both regimes.* ***Low-clouds resembling typical open and closed-cellular structures for stratocumulus are common in this location (Jensen et. al. 2021) and are found to form under the influence of the Azores high pressure system as well as behind frontal systems and cold-air outbreaks occurring during northwesterly flows. Low-clouds over the Azores frequently produce precipitation, mostly in the form of drizzle. The atmosphere is moister and less stable during winter than during summer, resulting in thicker cloud layers with higher LWP and drizzle frequency (~70%) during winter months (Rémillard et al., 2012, Wood et al., 2015, Wu et al., 2020). Using a satellite cloud regime analysis, Rémillard and Tselioudis (2015) show that the variability in the cloud distribution at the Azores is similar to the global mean, thereby making it an ideal location to evaluate aerosol and warm cloud properties and processes in climate model simulations.***

It is difficult to find a clear reason to reduce Nd dependency and increase Qc dependency for autoconversion constants. It will be helpful to describe that in Section 2.4. Same as for autoconversion parameters, the reason for the accretion factor change could not be found. Also, it is good to explain how the specific parameter values in Table 2 were selected.
>> The following has been added to Section 2.4. Please see Figures R1 and R2 for more details on the autoconversion rate calculations. In general, rates were modified being still realistic but perturbed enough to produce changes in ACI.
***We reduce dependency of autoconversion to $N_d$ and increase the $Q_c$ dependency to enhance the autoconversion rate. This leads to an approximate 15-30% increase in the in-cloud wet deposition rates (see Figure R1 and Table R1 above) in the A1-based experiments relative to A0. The equation for accretion rate is now provided in Section 2.4 and is varied by 50% depending on the experiment.***

Explains about Nd derivation from the CERES and MODIS data are missing.
>> We now provide a more thorough description of the computation of Nd and LWP in the manuscript. The following sentence was added to clarify how Nd is computed for each dataset.
***We use the equivalent form of $N_d = \gamma L^{5/6} N_d^{1/3}$, where γ = 1.37 × 10⁻⁵m⁻⁰·⁵ to compute cloud droplet number concentration from cloud effective radius and optical depth variables retrieved from ARM, satellite (MODIS and CERES) and obtained using***

*COSP in E3SM simulations. Grosvenor et al. (2018) can be consulted for a comprehensive assessment of the Nd uncertainties.*

For CERES data, I am concerned about using the dataset during the less trustworthy time (Figure 7 and Table 4, 5, 6, and 7). Figure 7 Re and Nd figures show significantly different values between trustworthy and less trustworthy time. I hope that the authors discuss this.
>> Because CERES cloud effective radius, optical thickness and liquid water path are not trustworthy at night-time, these data have been removed entirely from Figure 7 to avoid confusion. We have clarified that only "daytime averages" were used in Tables 4,5,6, and 7. This was true in the original submission so these estimates have not changed in the table but we have added clarifying text to the tables, specifically *"…the daylight period (roughly from 9 am - 3 pm local time)…"*.

I understood that this paper considers low-level clouds. If this is correct, I guess that 500 hPa could be too high to check the large-scale vertical motion affecting low-level clouds. I think this confusion could be automatically resolved if comment 1 is considered.
>> We use 500 hPa as an upper limit height for cloud tops, very few (less than 1%) that are still in liquid warm phase actually achieve such altitudes, the vast majority are below 700 hPa. We simply use Omega_500 hPa to study warm cloud regimes with large-scale ascent vs decent.

**Technical comments**
I would like to recommend using the words "CCN" and "CCN number concentration" more precisely (one example, "increase in CCN concentration" seems better in line 24). Because, as authors already know, concentration is not the only CCN property that could influence cloud (e.g., radius, hygroscopicity, …).
>> Thank you for pointing this out. I have added the word "concentration" in several places where "CCN" occurs by its own in the manuscript.

Both "MERRA2" and "MERRA-2" are used. It seems better to choose just one of them.
>> All instances of "MERRA2" have been changed to "MERRA-2" to be consistent with the literature.

line 25: constant changes -> no change >> Done

line 42: geophysical -> physical >> Done

line 89-90: complete a complete cycle -> complete one cycle >> Done

line 153: aerosol optical depth (AOD): AOD is already mentioned in line 64  >> Changed accordingly.

line 188: oceanic -> marine >> Done

line 220: start -> center >> To avoid confusion, it has been clarified here that it is the "start time" of the trajectory.

line 306: 3b -> 3d >> There is no Figure 3d. Figure 3b, referring to E3SM, in this sentence is correct.

line 358: 500-hPa vertical velocity, free tropospheric vertical velocity -> Does it mean "500-hPa free tropospheric vertical velocity"? >> We have modified it to read as "500-hPa free tropospheric vertical velocity"

line 393: The change -> The difference >> Done

line 398: from (Chen et al., 2014) -> from Chen et al. (2014) >> This error has been corrected.

line 446: global-based observational -> global observational-based >> Done

line 450: lagrangian -> Lagrangian  >> Done

line 497 and 506: equation 5 -> equation (5) >> Done

Figure 5a caption: rain rate -> rain rate (blue) or CCN concentration (red) -> CCN concentration >> To avoid confusion we have removed the "(red)"

Figure 5c caption: Is Figure 5c for A1R0 simulation? If so, why A1R0 simulation result is displayed here instead of the control simulation (A0R0)? >> This was a typo, it is actually A0R0.

Figure 7: description for c and d should be checked. >> Great catch! (c) has been changed to droplet number concentration and (d) has been changed to liquid water path.

Table 1: geophysical -> physical >> Done

Figure S4: Is this figure for relative frequency or precipitation rate? >> Another great catch! The figure captions S4 and S5 were in switched order and have now been corrected.

Figure S9a and b: W/m-2 -> W/m2 or W m-2 >> Units for sensible and latent heat fluxes have been corrected. I also noticed that the units for lifted condensation level were "km" when they should have been "m" and this has been corrected as well.

Figure S10: Description for pink shading is missing. >> Caption has been updated to include "pink regions denote the 5-95 % confidence interval.

Figure S14: LWP unit should be checked. >> Units for LWP have been corrected.

**References**

Abdul-Razzak, H., & Ghan, S. J. (2000). A parameterization of aerosol activation 2. Multiple aerosol types. *Journal of Geophysical Research-Atmospheres*, *105*(D5), 6837-6844. https://doi.org/Doi 10.1029/1999jd901161

Ebmeier, S. K., Sayer, A. M., Grainger, R. G., Mather, T. A., and Carboni, E.: Systematic Satellite Observations of the Impact of Aerosols from Passive Volcanic Degassing on Local Cloud Properties, Atmos. Chem. Phys., 14, 10 601–10 618, https://doi.org/10.5194/acp-14-10601-2014, 2014.

Ghan, S. J.: Technical Note: Estimating aerosol effects on cloud radiative forcing, Atmospheric Chemistry and Physics, 13, 9971–9974, https://doi.org/10.5194/acp-13-9971-2013, 2013.

Gettelman, A. and Morrison, H.: Advanced Two-Moment Bulk Microphysics for Global Models. Part I: Off-Line Tests and Comparison with Other Schemes, Journal of Climate, 28, 1268 – 1287, https://doi.org/10.1175/JCLI-D-14-00102.1, 2015

Grosvenor, D. P., et al.: Remote Sensing of Droplet Number Concentration in Warm Clouds: A Review of the Current State of Knowledge and Perspectives, Reviews of Geophysics, 56, 409–453, https://doi.org/10.1029/2017RG000593, 2018

Haarsma, R. J., et al.: High Resolution Model Intercomparison Project (HighResMIP v1.0) for CMIP6, Geoscientific Model Development, 9, 4185–4208, https://doi.org/10.5194/gmd-9-4185-2016, 2016

Jensen, M. P., Ghate, V. P., Wang, D., Apoznanski, D. K., Bartholomew, M. J., Giangrande, S. E., Johnson, K. L., and Thieman, M. M.: Contrasting characteristics of open- and closed-cellular stratocumulus cloud in the eastern North Atlantic, Atmospheric Chemistry and Physics, 21, 14 557–14 571, https://doi.org/10.5194/acp-21-14557-2021, 2021

Liu, X., Penner, J. E., Ghan, S. J., & Wang, M. (2007). Inclusion of ice microphysics in the NCAR community atmospheric model version 3 (CAM3). *Journal of Climate*, *20*(18), 4526-4547. https://doi.org/10.1175/Jcli4264.1

Rasch, P. J., Barth, M. C., Kiehl, J. T., Schwartz, S. E., & Benkovitz, C. M. (2000). A description of the global sulfur cycle and its controlling processes in the National Center for Atmospheric Research Community Climate Model, Version 3 (vol 105, pg 1367, 2000). *Journal of Geophysical Research-Atmospheres*, *105*(D5), 6783-6783. https://doi.org/Doi 10.1029/2000jd900036

Rasch, P. J., et al. : An Overview of the Atmospheric Component of the Energy Exascale Earth System Model, Journal of Advances in Modeling Earth Systems, 11, 2377–2411, https://doi.org/https://doi.org/10.1029/2019MS001629, 2019

Rémillard, J. and Tselioudis, G.: Cloud Regime Variability over the Azores and Its Application to Climate Model Evaluation, Journal of Climate, 28, 9707 – 9720, https://doi.org/10.1175/JCLI-D-15-0066.1, 2015.

Rémillard, J., Kollias, P., Luke, E., and Wood, R.: Marine Boundary Layer Cloud Observations in the Azores, Journal of Climate, 25, 7381– 7398, https://doi.org/10.1175/JCLI-D-11-00610.1, 2012.

Tang, Q., et al.: Regionally refined test bed in E3SM atmosphere model version 1 (EAMv1) and applications for high-resolution modeling, Geoscientific Model Development, 12, 2679–2706, https://doi.org/10.5194/gmd-12-2679-2019, 2019

Wang, H., et al.: Sensitivity of remote aerosol distributions to representation of cloud–aerosol interactions in a global climate model, Geoscientific Model Development, 6, 765–782, https://doi.org/10.5194/gmd-6-765-2013, 2013.

Wang, Y., Liu, X., Hoose, C., & Wang, B. (2014). Different contact angle distributions for heterogeneous ice nucleation in the Community Atmospheric Model version 5. *Atmospheric Chemistry and Physics*, *14*(19), 10411-10430. https://doi.org/10.5194/acp-14-10411-2014

Wood, R., et al.: CLOUDS, AEROSOLS, AND PRECIPITATION IN THE MARINE BOUNDARY LAYER: An ARM Mobile Facility Deployment, Bulletin of the American Meteorological Society, 96, 419–440, http://www.jstor.org/stable/26219584, 2015

Wu, P., Dong, X. Q., Xi, B. K., Tian, J. J., and Ward, D. M.: Profiles of MBL Cloud and Drizzle Microphysical Properties Retrieved From Ground-Based Observations and Validated by Aircraft In Situ Measurements Over the Azores, Journal of Geophysical Research-Atmospheres, 125, https://doi.org/ARTN e2019JD032205 10.1029/2019JD032205, 2020

Xie, S., et al.: Understanding Cloud and Convective Characteristics in Version 1 of the E3SM Atmosphere Model, Journal of Advances in Modeling Earth Systems, 10, 2618–2644,945, https://doi.org/https://doi.org/10.1029/2018MS001350, 2018